

# Carroll expansion of general relativity

Dennis Hansen[1], Niels A. Obers[2,3], Gerben Oling[2,3] and Benjamin T. Søgaard[3,4]

**1** Institut für Theoretische Physik, Eidgenössische Technische Hochschule Zürich,
Wolfgang-Pauli-Strasse 27, 8093 Zürich, Switzerland
**2** Nordita, KTH Royal Institute of Technology and Stockholm University,
Hannes Alfvéns väg 12, SE-106 91 Stockholm, Sweden
**3** The Niels Bohr Institute, University of Copenhagen, Blegdamsvej 17,
DK-2100 Copenhagen Ø, Denmark
**4** Department of Physics, Princeton University, Princeton, NJ 08544, USA

## Abstract

We study the small speed of light expansion of general relativity, utilizing the modern perspective on non-Lorentzian geometry. This is an expansion around the ultra-local Carroll limit, in which light cones close up. To this end, we first rewrite the Einstein–Hilbert action in pre-ultra-local variables, which is closely related to the 3+1 decomposition of general relativity. At leading order in the expansion, these pre-ultra-local variables yield Carroll geometry and the resulting action describes the electric Carroll limit of general relativity. We also obtain the next-to-leading order action in terms of Carroll geometry and next-to-leading order geometric fields. The leading order theory yields constraint and evolution equations, and we can solve the evolution analytically. We furthermore construct a Carroll version of Bowen–York initial data, which has associated conserved boundary linear and angular momentum charges. The notion of mass is not present at leading order and only enters at next-to-leading order. This is illustrated by considering a particular truncation of the next-to-leading order action, corresponding to the magnetic Carroll limit, where we find a solution that describes the Carroll limit of a Schwarzschild black hole. Finally, we comment on how a cosmological constant can be incorporated in our analysis.



# 1 Introduction

The theory of General Relativity (GR) beautifully describes the dynamics of space and time, incorporating local Lorentz symmetry through Einstein's equivalence principle. Lorentz boosts are transformations that depend on the speed of light $c$. An obvious and important regime to study is the one in which $c$ is very large. In concrete physical setups, such a limit results in non-relativistic systems with Galilean symmetry, where the appropriate characteristic velocity is much less than the speed of light. This familiar and well-studied limit is relevant for post-Newtonian (PN) approximations of solutions of GR, with many important applications in astrophysics and cosmology [1–3].

In a recent development, a geometric description of the non-relativistic expansion of GR in inverse powers of the speed of light $c$ was obtained in Refs. [4–6], building on earlier work [7]. This progress was spurred on in large part by novel insights [8–10] into Newton–Cartan geometry, which is the geometry that replaces the Lorentzian geometry of GR at leading order in the expansion. The resulting action for non-relativistic gravity (which is the next-to-next-to leading order action in the expansion) includes Newtonian gravity, but also goes beyond it by allowing for strong gravitational fields, which leads for example to gravitational time dilation. More generally, these expansion methods lay the foundation for a covariant and off-shell formulation to any order and are therefore expected to be relevant to PN expansions. Similar non-relativistic geometries, along with corresponding probe and spacetime actions, also prominently appear in modern approaches to non-relativistic field theory and string theory (see for example [11–17]).

It is also interesting to consider the opposite case, namely the regime in which $c$ is very small. In particular, as first considered in Ref. [18–20], the Poincaré group contracts to the

Carroll group when the speed of light is taken to zero. This has some unusual consequences for the kinematics and dynamics in this limit. Contrary to the Galilean case, where light cones open up, the light cones close up in the Carroll limit. Particles with non-zero energy cannot move in space anymore, and for these particles there can be no interactions between spatially separated events.[1] Hence this Carroll limit is an ultra-local limit and its study in GR, which goes back to [23], provides novel insights into its geometry and gravitational dynamics. More generally, studying the small speed of light expansion of GR [24] allows for a perturbative expansion around the (singular) Carroll point, analogous to the PN expansion for large $c$.

In this paper, we will perform a systematic analysis of the ultra-local Carroll expansion of GR, utilizing the modern perspective on non-Lorentzian geometry, in analogy with the recent results [4–6] on the non-relativistic expansion. Compared to the latter, the Carroll expansion of general relativity will encode different aspects of the full Lorentzian theory at each order. On a technical level, the Carroll expansion may help us to develop novel analytical approaches which can subsequently be applied in the non-relativistic expansion, whose astrophysical relevance is more obvious. In fact, we will see that the structure of the Carroll expansion clearly brings out an analogy to the 3+1 formulation of GR and the resulting Hamiltonian and momentum constraints.

Furthermore, the dynamics of GR that are captured by the Carroll expansion may themselves be of physical relevance, too. Contrary to the large speed of light expansion of GR, there is already non-trivial dynamics at leading order in the Carroll expansion. In particular, while the 'kinetic' term containing extrinsic curvatures appears at next-to-next-to-leading order in the former case, it already appears at leading order in the latter. The resulting dynamics appears to be closely related to the Beliniski-Khalatnikov-Lifshitz limit [25], which describes the near-singularity dynamics of general relativity. Although we will not pursue this connection in the present work, the techniques we develop could allow us to explore subleading orders of this limit.

In order to perform the expansion, we will first reformulate the Einstein–Hilbert (EH) action in terms of a 'pre-ultra-local' (PUL) variables, which is similar to the 3+1 decomposition. This way of writing the EH action is related to the 'pre-non-relativistic' (PNR) formulation [5, 6], though the connection is different and the geometric variables that are used are in some sense dual to each other. This duality also manifests itself when expanding: the large $c$ expansion of the PNR variables leads to Newton–Cartan geometry plus higher-order geometric fields, while the small $c$ expansion of the PUL variables leads to Carroll geometry plus higher-order geometric fields. At leading order, this observation reduces to the well-known duality between NC and Carroll geometry [26, 27].

More generally, physics with Carroll symmetries has recently appeared in a wide variety of studies, including various aspects of Carroll gravity and geometry [21, 26–50]. In particular, since null hypersurfaces are examples of Carroll manifolds [26], Carroll symmetries have been related to black holes [34, 35], while the recent Ref. [21] considers Carroll symmetry in relation to inflation and cosmology.[2] Furthermore, Carroll field theories have been studied for example in [21, 26, 47, 53–60]. Carroll symmetry also features in 3D flat space holography and tensionless strings [53, 61–69], while Carroll fluids have been addressed in [70–73].

The approach pursued in the present paper, considering Carroll expansions instead of taking the Carroll limit, was also put forward in [21], where it was motivated by applications to cosmology, inflation and dark energy. In line with the philosophy advocated earlier for GR in [24], the idea is that the Carroll symmetry of the limiting point $c = 0$ of a theory provides

---

[1]There is a second type of Carroll particle [21], with zero energy, which cannot stand still, and originates from the Carroll limit of relativistic tachyons. See also [21, 22] for non-trivial dynamics of coupled Carroll particles.

[2]See also Ref. [51] for superluminal behavior in relation to cosmology for a brane universe moving in a curved higher dimensional bulk space and Ref. [52] for connections between the ultra-local limit of gravity and cosmological billiards.

an organizing principle in the study of the perturbative expansion around it. To illustrate this, the Carroll expansion of the action of a scalar field and Maxwell theory were discussed in [21]. Furthermore, it was shown that the expansions can be truncated in such a way that that these relativistic theories admit two inequivalent contractions with Carroll invariance. This feature follows from the observation [21] that not only is the leading-order (LO) action manifestly Carroll invariant, but one can also obtain a Carroll-invariant theory if one takes the next-to-leading (NLO) action and restricts it to the field configurations for which the LO action vanishes.

The fact that relativistic theories generically admit two inequivalent contractions with Carroll symmetry was shown independently from a Hamiltonian perspective for general $p$-form theories in [47], where it was also briefly considered for GR. Following these authors, we will refer to the two distinct contractions for a given theory as the 'electric' and 'magnetic' theory, respectively. These names are derived from the two possible contractions that were already known for electromagnetism [26], though the magnetic theory was only known at the level of the equations of motion. As far as GR is concerned, the electric contraction of GR has been known since many years [23], but the existence of the magnetic contraction has only recently been realized [47, 48]. In the present work, following the suggestion of [21], we will use the Lagrangian perspective to obtain for the first time a covariant action for the magnetic Carroll sector of gravity. However, it is important to stress that both the electric and magnetic Carroll limits are special cases of the general framework we develop, corresponding to the LO theory and a truncation of the NLO theory, respectively. While these turn out to be interesting and tractable subsectors, our general Carroll expansion will be able to describe much more general dynamics at subleading orders.

As it turns out, the Carroll LO action already contains non-trivial equations of motion. These can be written in a form that is very similar to the constraint and evolution equations in the 3+1 decomposition of GR. In the full relativistic theory, solving the evolution equations even numerically is a formidable problem. In contrast, given initial data satisfying the LO constraint equations, its evolution can be solved analytically. Building on methods from the 3+1 decomposition of GR, we construct a Carroll version of Bowen–York initial data [74], which describes black holes parametrized by their mass and angular/linear momentum in the full relativistic theory. We subsequently show that the parameters in the resulting LO solution correspond to conserved boundary charges associated to asymptotic translations and rotations. Although the notion of mass or energy is not yet present in the LO theory [48], it is interesting that it still contains such a class of physically relevant solutions. To obtain a mass, one should consider the NLO theory. To illustrate this, we consider for simplicity the subsector comprised by the magnetic theory described above. (Here, an asymptotic symmetry analysis was previously carried out in [48].) Focusing on static solutions, we find a Carroll geometry that describes the ultra-local limit of a Schwarzschild black hole in isotropic coordinates.

We also comment on how a cosmological constant can be incorporated in the Carroll expansion of GR, and we discuss some solutions for the electric (LO) and magnetic (truncated NLO) theories. The relevance of the cosmological constant at these orders depends on its scaling in terms of powers of $c$. For the LO theory we find a Carroll solution corresponding to the ultra-local limit of de Sitter geometry. This Carroll geometry was also considered in [21] and was earlier found as a homogeneous background in [75], where it was called the 'light cone'. Interestingly, while a negative cosmological constant is inconsistent with the equations of motion of the electric theory, we will show that both signs are allowed in the magnetic theory.

At this point, a comment is in order on the meaning of expanding GR in terms of a small (or large) speed of light $c$, which is after all a dimensionful parameter. In a given physical problem, we will have a characteristic velocity $v_c$ that allows us to expand in $c/v_c$, which is

dimensionless. Clearly, such an expansion breaks some form of general covariance, since the characteristic velocity refers to a particular set of frames (or coordinate system). In the following, instead of considering a particular relativistic problem with a particular characteristic velocity, we will mainly focus on developing the general ultra-local small $c$ expansion of general relativity. Alternatively, this can be interpreted as setting $c = \hat{c}\,\epsilon$, where $\epsilon$ is dimensionless, and expanding $\epsilon$ around zero.

Finally, it is important to note that the meaning of a 'small $c$' limit depends on how the dimensionless parameter $c/v_c$ behaves in this limit [21]. If we take $c/v_c \to 0$, the characteristic velocity tends to zero slower than the speed of light, and hence all dynamics should freeze out. For this reason, we refer to the limit $c/v_c \to 0$ as the ultra-local limit. On the other hand, if we take $c/v_c \to 1$, the characteristic velocity tends to the speed of light. The latter corresponds to the more common notion of an ultra-relativistic limit, which we do not consider here.

This paper is organized as follows. In Section 2, we introduce the main concepts of Carroll geometry using a small speed of light expansion of Lorentzian geometry. In particular, we show how local Carroll boosts are obtained from local Lorentz boosts, and we identify a suitable connection and show how to relate it and its curvature to the Levi-Civita connection and its curvature. We also discuss the notion of spatial hypersurfaces and their induced connection and curvature in the context of Carroll geometry. Next, in Section 3, we develop the small speed of light expansion of Einstein gravity. We identify the resulting leading-order action and a subsector of the subleading Lagrangians with the 'electric' and 'magnetic' theories that were recently formulated from a Hamiltonian perspective, we derive their equations of motion and we separate them into evolution and constraint equations. In particular, we obtain a manifestly covariant action for the magnetic theory, which was unknown from the Hamiltonian perspective, and we show how it can be obtained from a limit of the Einstein-Hilbert action. In Section 4 we then show that the evolution equations of the electric theory can be solved analytically. We find solutions to the constraint equations with angular and linear momentum, which we verify using boundary charges that we construct the covariant phase space formalism. In the subleading magnetic theory, we identify Schwarzschild-like solutions with non-zero mass. Finally, we show how a cosmological constant can be included in both theories. We conclude this paper in Section 5, where we summarize our results and list future directions. Appendix A lists our conventions and provides some useful identities involving our Carroll connection, while Appendix B provides an alternative derivation of this connection using frame bundles.

## 2 Carroll geometry from Lorentzian geometry

In this section, we develop the small speed of light expansion of the Lorentzian geometry of general relativity. To this end, we first introduce a *'pre-ultra-local'* (PUL) parametrization of the Lorentzian metric and vielbeine. This parametrization is adapted to the ultra-local structure that arises in the expansion. To be precise, at leading order, the expansion of the PUL variables leads to a notion of *Carroll geometry*. In this geometry, the light cone of Lorentzian geometry collapses to a line, and we will see that the resulting leading-order theory of gravity exhibits ultra-local behavior.

Furthermore, the corresponding vielbeine transform under local Carroll boosts instead of local Lorentz boosts. As a result, in the context of the ultra-local expansion, the usual Levi-Civita connection of Lorentzian geometry is no longer a natural choice. Instead, we introduce a convenient Carroll-compatible connection and show how it can be obtained from a PUL parametrization of the Levi-Civita connection. Subsequently, we determine the relation between the Levi-Civita curvature that enters in the Einstein–Hilbert action and the curvature of

our PUL connection.

With this, we obtain a PUL reparametrization of the Lorentzian geometric variables of general relativity that is adapted to the emergence of Carroll symmetry in the ultra-local expansion. The results of this section are closely related to the 'pre-non-relativistic' (PNR) parametrization that was used for the non-relativistic expansion of GR in Section 2 of [6], with the main difference between the two being the choice of connection.

## 2.1 'Pre-ultra-local' parametrization and expansion

In terms of the Lorentzian metric $g_{\mu\nu}$ and its inverse $g^{\mu\nu}$, the starting point for the 'ultra-local' small speed of light expansion we consider is the PUL parametrization

$$g_{\mu\nu} = -c^2 T_\mu T_\nu + \Pi_{\mu\nu}, \qquad g^{\mu\nu} = -\frac{1}{c^2} V^\mu V^\nu + \Pi^{\mu\nu}. \qquad (2.1)$$

In this parametrization, we have introduced a 'timelike' one-form and vector $T_\mu$ and $V^\mu$ as well as the 'spatial' symmetric tensors $\Pi_{\mu\nu}$ and $\Pi^{\mu\nu}$. These PUL variables satisfy the following orthonormality and completeness relations,

$$T_\mu V^\mu = -1, \quad T_\mu \Pi^{\mu\nu} = 0, \quad \Pi_{\mu\nu} V^\nu = 0, \quad \delta^\mu_\nu = -V^\mu T_\nu + \Pi^{\mu\rho}\Pi_{\rho\nu}. \qquad (2.2)$$

Roughly speaking, the PUL parametrization corresponds to a split of the tangent bundle in 'temporal' and 'spatial' components, as we discuss in more detail in Section 2.5. Additionally, it makes the factors of $c^2$ that appear in the Lorentzian metric explicit, so that the resulting variables can be expanded uniformly in the speed of light.

In terms of the (inverse) vielbeine $E_\mu{}^A$ and $\Theta^\nu{}_A$, which are related to the metric using $g_{\mu\nu} = \eta_{AB} E_\mu{}^A E_\nu{}^B$ and $g^{\mu\nu} = \eta^{AB}\Theta^\mu{}_A\Theta^\nu{}_B$, this parametrization corresponds to

$$E_\mu{}^A = \left( c\, T_\mu, E_\mu{}^a \right), \qquad \Theta^\mu{}_A = \left( -\frac{1}{c} V^\mu, \Theta^\mu{}_a \right), \qquad (2.3)$$

so that $\Pi_{\mu\nu} = \delta_{ab} E_\mu{}^a E_\nu{}^b$ and $\Pi^{\mu\nu} = \delta^{ab}\Theta^\mu{}_a\Theta^\nu{}_b$. Here, $A = 0, 1, \ldots, d$ are spacetime frame indices and $a = 1, \ldots, d$ are spatial frame indices.

Next, assuming that the resulting PUL vielbeine (2.3) are analytic in $c^2$, we can expand them as follows,

$$V^\mu = v^\mu + c^2 M^\mu + \mathcal{O}(c^4), \qquad\qquad T_\mu = \tau_\mu + \mathcal{O}(c^2), \qquad (2.4a)$$

$$\Theta^\mu{}_a = \theta^\mu{}_a + c^2 \pi^\mu{}_a + \mathcal{O}(c^4), \qquad\qquad \mathcal{E}_\mu{}^a = e_\mu{}^a + \mathcal{O}(c^2), \qquad (2.4b)$$

$$\Pi^{\mu\nu} = h^{\mu\nu} + c^2 \Phi^{\mu\nu} + \mathcal{O}(c^4), \qquad\qquad \Pi_{\mu\nu} = h_{\mu\nu} + \mathcal{O}(c^2), \qquad (2.4c)$$

where $h_{\mu\nu} = \delta_{ab} e_\mu{}^a e_\nu{}^b$, $h^{\mu\nu} = \delta^{ab}\theta^\mu{}_a\theta^\nu{}_b$ and $\Phi^{\mu\nu} = \delta^{ab}(\theta^\mu{}_a\pi^\nu{}_b + \pi^\mu{}_a\theta^\nu{}_b)$.

It is important to stress the limitations and the physical implications of the assumptions that allow us to write down the expansion (2.4). First, following the Galilean expansion developed in Section 2.1 of [6], we restrict ourselves here to an expansion in even powers. If the metric/vielbeine are not analytic in $c^2$, the corresponding coordinate system/frame is not suitable for the expansion we describe here, but see [76] for the inclusion of odd powers in the non-relativistic expansion. More generally, the expansion (2.3) is clearly not preserved by any coordinate transformations that are not analytic in $c^2$, so that for example Schwarzschild and Kruskal–Szekeres coordinates correspond to inequivalent starting points. In line with the discussion at the end of the Introduction, the Carroll expansion of a Lorentzian geometry depends on a choice of time coordinate, and different choices will allow us to probe different aspects of the full relativistic geometry.

Next, note that we can solve the variables appearing at subleading orders in the expansion of $T_\mu$, $\mathcal{E}_\mu{}^a$ and $\Pi_{\mu\nu}$ in (2.4) in terms of the variables in the expansion of $V^\mu$, $\Theta^\mu{}_a$ and $\Pi^{\mu\nu}$ using the relations (2.2), so we will not introduce separate variables for them. The leading-order terms then satisfy

$$\tau_\mu v^\mu = -1, \quad \tau_\mu h^{\mu\nu} = 0, \quad h_{\mu\nu} v^\nu = 0, \quad \delta^\mu_\nu = -v^\mu \tau_\nu + h^{\mu\rho} h_{\rho\nu}. \tag{2.5}$$

As we will show in the following, these leading-order variables define a Carroll geometry. In particular, we now demonstrate that their local symmetries (corresponding to Carroll transformations) can be obtained from the small $c$ expansion of local Lorentz transformations. Subsequently, we introduce a convenient Carroll-compatible connection.

## 2.2 Carroll boosts from Lorentz boosts

In general relativity, the Lorentzian vielbeine transform under the local Lorentz transformations $\Lambda^A{}_B$ that preserve the Minkowski metric $\eta_{AB}$ on the frame bundle,

$$\delta_\Lambda E^A = \Lambda^A{}_B E^B, \quad \delta_\Lambda \Theta_A = -\Lambda^B{}_A \Theta_B, \qquad \Lambda^{AB} = -\Lambda^{BA}. \tag{2.6}$$

Indices are raised and lowered using $\eta_{AB}$ and its inverse. These Lorentz transformations correspond to the following transformations of the PUL vielbeine (2.3),

$$\delta_\Lambda T_\mu = \Lambda_a \mathcal{E}_\mu{}^a, \qquad\qquad \delta_\Lambda \mathcal{E}_\mu{}^a = c^2 \Lambda^a T_\mu + \Lambda^a{}_b \mathcal{E}_\mu{}^b, \tag{2.7a}$$

$$\delta_\Lambda V^\mu = c^2 \Lambda^a \Theta^\mu{}_a, \qquad\qquad \delta_\Lambda \Theta^\mu{}_a = \Lambda_a V^\mu - \Lambda^b{}_a \Theta^\mu{}_b. \tag{2.7b}$$

Here, we have introduced the rescaled generators $\Lambda_a = c\Lambda^0{}_a$, which we take to be finite in the $c \to 0$ limit. Next, we expand the rotation and boost parameters using

$$\Lambda^a{}_b = \lambda^a{}_b + \mathcal{O}(c^2), \qquad \Lambda_a = \lambda_a + \mathcal{O}(c^2). \tag{2.8}$$

The leading-order Carroll vielbeine coming from the expansion (2.4) then transform as

$$\delta_\lambda \tau_\mu = \lambda_a e_\mu{}^a, \quad \delta_\lambda e_\mu{}^a = \lambda^a{}_b e_\mu{}^b, \qquad \delta_\lambda v^\mu = 0, \quad \delta_\lambda \theta^\mu{}_a = \lambda_a v^\mu - \lambda^b{}_a \theta^\mu{}_b. \tag{2.9}$$

The parameters $\lambda^a{}_b$ correspond to spatial rotations, while $\lambda_a$ generate Carroll boosts, and their indices are raised and lowered using the Kronecker delta. We see that only the timelike vector $v^\mu$ and the spatial metric $h_{\mu\nu} = \delta_{ab} e_\mu{}^a e_\nu{}^b$ are invariant under boosts, while $\tau_\mu$ and $h^{\mu\nu} = \delta^{ab} \theta^\mu{}_a \theta^\nu{}_b$ transform under boosts. In particular, we have

$$\delta_\lambda h^{\mu\nu} = 2\lambda^{(\mu} v^{\nu)}, \qquad \lambda^\mu = h^{\mu\nu} \lambda_\nu = \theta^\mu{}_a \lambda^a. \tag{2.10}$$

Note that the fact that $\tau_\mu$ transforms under boosts while $v^\mu$ does not is compatible with the fact that $\tau_\mu v^\mu = -1$, since the variation of the latter is $\lambda_a e_\mu{}^a v^\mu = 0$.

These local Carroll transformations arise from the $c \to 0$ limit of the local Lorentz transformations, and they are a fundamental part of the resulting Carroll geometry. Note that, while individual tensors such as $h^{\mu\nu}$ may transform under Carroll boosts, the combinations that arise from limits of Lorentz-boost invariant quantities will be Carroll-boost invariant. Finally, we remark that the Carroll vielbeine and their transformations can also be obtained by gauging the Carroll algebra [27].

At higher orders in $c^2$, additional vielbein variables and additional local symmetry transformations arise. In the non-relativistic expansion, the leading-order fields describe Newton–Cartan geometry with local Galilean symmetry. Including the fields at next-to-leading order defines what is known as type II torsional Newton–Cartan geometry (see Section 2 of [6]). Likewise, the additional variables and transformations that arise at higher orders in the ultra-local expansion can be interpreted as additional fields. In the following, we will mainly be concerned with the leading-order Carroll geometry, and we also explore parts of its next-to-leading-order corrections. However, the systematics of our ultra-local expansion can in principle be extended to any order.

## 2.3 Carroll-compatible connection

As we have seen in the above, Carroll metric variables arise from the leading order of the ultra-local expansion of a Lorentzian metric. We now introduce a convenient connection that is compatible with the Carroll variables.

In Equation (2.9), we saw that the timelike vector $v^\mu$ and the spatial metric $h_{\mu\nu}$ are invariant under local Carroll boosts, but their inverses $\tau_\mu$ and $h^{\mu\nu}$ are not. For this reason, the requirement that the covariant derivative of $\tau_\mu$ and $h^{\mu\nu}$ vanishes would not be a boost-invariant statement in general Carroll backgrounds. It is therefore more appropriate to work with a connection $\tilde{\Gamma}^\rho_{\mu\nu}$ such that only $v^\mu$ and $h_{\mu\nu}$ are covariantly constant with respect to the associated covariant derivative,

$$\tilde{\nabla}_\mu v^\nu = 0, \qquad \tilde{\nabla}_\rho h_{\mu\nu} = 0. \tag{2.11}$$

As we will see shortly, such a connection typically has non-zero torsion, and therefore it cannot arise out of an expansion of the Levi-Civita connection that is commonly used in Lorentzian geometry. One way to obtain a connection satisfying (2.11) at leading order is to introduce a different connection $\tilde{C}^\rho_{\mu\nu}$ which satisfies similar requirements in terms of the PUL variables already *before* the ultra-local expansion,

$$\overset{(c)}{\nabla}_\mu V^\nu = 0, \qquad \overset{(c)}{\nabla}_\rho \Pi_{\mu\nu} = 0. \tag{2.12}$$

Such a connection is not particularly natural from the point of view of Lorentzian geometry, since the PUL vielbeine still transform under the usual Lorentz boosts (2.6). Instead, it serves to accommodate the Carroll structure that arises as a result of the expansion. This mirrors the Galilean-compatible PNR connection that was considered for the non-relativistic expansion in Section 2.1 of [6], see also [77] for a discussion from the perspective of the first-order formulation of gravity.

The conditions (2.12) do not fully determine the connection. As we discuss in Appendix B, a convenient choice is given by

$$\tilde{C}^\rho_{\mu\nu} = -V^\rho \partial_{(\mu} T_{\nu)} - V^\rho T_{(\mu} \mathcal{L}_V T_{\nu)} + \frac{1}{2} \Pi^{\rho\lambda} \left[ \partial_\mu \Pi_{\nu\lambda} + \partial_\nu \Pi_{\lambda\mu} - \partial_\lambda \Pi_{\mu\nu} \right] - \Pi^{\rho\lambda} T_\nu \mathcal{K}_{\mu\lambda}. \tag{2.13}$$

Here, $\mathcal{K}_{\mu\nu} = -\frac{1}{2} \mathcal{L}_V \Pi_{\mu\nu}$ is the extrinsic curvature. Note that this is a symmetric and purely spatial tensor, since $V^\mu \mathcal{K}_{\mu\nu} = 0$. At leading order, it leads to a Carroll-compatible connection satisfying (2.11),

$$\tilde{\Gamma}^\rho_{\mu\nu} = \tilde{C}^\rho_{\mu\nu}\Big|_{c=0} = -v^\rho \partial_{(\mu} \tau_{\nu)} - v^\rho \tau_{(\mu} \mathcal{L}_v \tau_{\nu)} + \frac{1}{2} h^{\rho\lambda} \left[ \partial_\mu h_{\nu\lambda} + \partial_\nu h_{\lambda\mu} - \partial_\lambda h_{\mu\nu} \right] - h^{\rho\lambda} \tau_\nu K_{\mu\lambda}, \tag{2.14}$$

where $K_{\mu\nu} = -\frac{1}{2} \mathcal{L}_v h_{\mu\nu}$ is the extrinsic curvature at leading order, which is also purely spatial, since it satisfies $v^\mu K_{\mu\nu} = 0$. This connection, which previously appeared in [29], is our preferred connection to describe the Carroll geometry that arises at leading order in the ultra-local expansion of general relativity. It is a special case of the general class of Carroll connections satisfying the compatibility requirements (2.11), which was determined in [27, 29].

In the next section, we will relate the PUL connection $\tilde{C}^\rho_{\mu\nu}$ and its curvature to the Levi-Civita connection and its curvature. For that, it is convenient to list some properties of the PUL connection. The equivalent expressions for the Carroll connection $\tilde{\Gamma}^\rho_{\mu\nu}$ can be easily obtained by expanding to leading order using (2.4). The torsion of the PUL connection is

$$2\tilde{C}^\rho_{[\mu\nu]} = 2\Pi^{\rho\lambda} T_{[\mu} \mathcal{K}_{\nu]\lambda}, \tag{2.15}$$

which is generically nonzero. The non-zero PUL metric covariant derivatives are

$$\overset{(c)}{\nabla}_\mu T_\nu = \frac{1}{2} T_{\mu\nu} - T_{(\mu} \mathcal{L}_V T_{\nu)} = \frac{1}{2} T_{\mu\nu} - V^\rho T_{\rho(\mu} T_{\nu)}, \tag{2.16a}$$

$$\overset{(c)}{\nabla}_\rho \Pi^{\mu\nu} = -V^{(\mu} \Pi^{\nu)\sigma} T_{\sigma\lambda} \left[ \delta_\rho^\lambda - V^\lambda T_\rho \right], \tag{2.16b}$$

where we have defined $T_{\mu\nu} = 2\partial_{[\mu} T_{\nu]}$. Note that the trace of the PUL connection is

$$\tilde{C}^\rho_{\rho\nu} = -V^\rho \partial_\nu T_\rho + \frac{1}{2} \Pi^{\rho\lambda} \partial_\nu \Pi_{\rho\lambda} - T_\nu \mathcal{K} = \frac{1}{E} \partial_\nu E - T_\nu \mathcal{K}, \tag{2.17}$$

where $E = \det(T_\mu, E_\mu{}^a)$ is the vielbein determinant and $\mathcal{K} = \Pi^{\mu\nu} \mathcal{K}_{\mu\nu}$ is the trace of the extrinsic curvature. This means that we can write a divergence containing this covariant derivative as a total derivative plus a term proportional to the extrinsic curvature,

$$\overset{(c)}{\nabla}_\mu X^\mu = \partial_\mu X^\mu + \tilde{C}^\rho_{\rho\nu} X^\nu = \frac{1}{E} \partial_\mu (E X^\mu) - \mathcal{K} T_\mu X^\mu. \tag{2.18}$$

This allows us to simplify such divergences inside a spacetime integral,

$$\int_M \overset{(c)}{\nabla}_\mu X^\mu E \, d^{d+1}x \approx -\int_M \mathcal{K} T_\mu X^\mu E \, d^{d+1}x, \tag{2.19}$$

which holds up to boundary terms (denoted using $\approx$). We will return to a careful study of such boundary terms and use them to define conserved charges in Section 4.2.

## 2.4 Decomposition of Levi-Civita connection and curvature

Our next goal is to show how our Carroll-compatible connection $\tilde{\Gamma}^\rho_{\mu\nu}$ defined in Equation (2.14) arises from the Levi-Civita connection of general relativity. First, using the PUL decomposition (2.1), the Levi-Civita connection $\Gamma^\rho_{\mu\nu}$ can be written as

$$\Gamma^\rho_{\mu\nu} = \frac{1}{2} g^{\rho\lambda} \left[ \partial_\mu g_{\nu\lambda} + \partial_\nu g_{\lambda\mu} - \partial_\lambda g_{\mu\nu} \right] \tag{2.20}$$

$$= \frac{1}{c^2} \overset{(-2)}{C}{}^\rho_{\mu\nu} + \tilde{C}^\rho_{\mu\nu} + S^\rho_{\mu\nu} + c^2 \overset{(2)}{C}{}^\rho_{\mu\nu}. \tag{2.21}$$

Note that we have not yet performed an expansion in $c^2$ yet. We have merely collected all terms that scale as $1/c^2$ or $c^2$ into the tensors $\overset{(-2)}{C}{}^\rho_{\mu\nu}$ and $\overset{(2)}{C}{}^\rho_{\mu\nu}$, respectively. Additionally, we have introduced a 'shift' tensor $S^\rho_{\mu\nu}$ that is tuned to produce our PUL connection $\tilde{C}^\rho_{\mu\nu}$ at level zero. These terms are given by

$$\overset{(-2)}{C}{}^\rho_{\mu\nu} = -V^\rho \mathcal{K}_{\mu\nu}, \tag{2.22a}$$

$$S^\rho_{\mu\nu} = \Pi^{\rho\lambda} T_\nu \mathcal{K}_{\mu\lambda}, \tag{2.22b}$$

$$\overset{(2)}{C}{}^\rho_{\mu\nu} = -T_{(\mu} \Pi^{\rho\lambda} (dT)_{\nu)\lambda}. \tag{2.22c}$$

Using this decomposition, any term involving the Levi-Civita covariant derivative can be rewritten into an expression involving the PUL covariant derivative.

In the following section, we will study the expansion of the action and the equations of motion of general relativity. To prepare for this, we now show how the Ricci tensor of the Levi-Civita connection can be decomposed in terms of PUL variables following (2.21),

$$R_{\mu\nu} = -\partial_\mu \Gamma^\rho_{\rho\nu} + \partial_\rho \Gamma^\rho_{\mu\nu} - \Gamma^\rho_{\mu\lambda} \Gamma^\lambda_{\rho\nu} + \Gamma^\rho_{\rho\lambda} \Gamma^\lambda_{\mu\nu} \tag{2.23}$$

$$= \frac{1}{c^4} \overset{(-4)}{R}{}_{\mu\nu} + \frac{1}{c^2} \overset{(-2)}{R}{}_{\mu\nu} + \overset{(0)}{R}{}_{\mu\nu} + c^2 \overset{(2)}{R}{}_{\mu\nu} + c^4 \overset{(4)}{R}{}_{\mu\nu}. \tag{2.24}$$

Again, we have only collected terms that scale equally in $c^2$, without expanding the individual tensors. The terms in this decomposition are given by

$$\overset{(-4)}{R}_{\mu\nu} = 0, \tag{2.25a}$$

$$\overset{(-2)}{R}_{\mu\nu} = \overset{(\tilde{C})}{\nabla}_\rho \overset{(-2)}{\tilde{C}}{}^\rho_{\mu\nu} - 2\tilde{C}^\lambda_{[\mu\rho]} \overset{(-2)}{\tilde{C}}{}^\rho_{\lambda\nu} - S^\rho_{\mu\lambda} \overset{(-2)}{\tilde{C}}{}^\lambda_{\rho\nu} + S^\rho_{\rho\lambda} \overset{(-2)}{\tilde{C}}{}^\lambda_{\mu\nu}, \tag{2.25b}$$

$$\overset{(0)}{R}_{\mu\nu} = \overset{(\tilde{C})}{R}_{\mu\nu} + \overset{(\tilde{C})}{\nabla}_\rho S^\rho_{\mu\nu} - \overset{(\tilde{C})}{\nabla}_\mu S^\rho_{\rho\nu} - 2\tilde{C}^\lambda_{[\mu\rho]} S^\rho_{\lambda\nu} - \overset{(-2)}{\tilde{C}}{}^\rho_{\mu\lambda} \overset{(2)}{\tilde{C}}{}^\lambda_{\rho\nu} - \overset{(2)}{\tilde{C}}{}^\rho_{\mu\lambda} \overset{(-2)}{\tilde{C}}{}^\lambda_{\rho\nu}, \tag{2.25c}$$

$$\overset{(2)}{R}_{\mu\nu} = \overset{(\tilde{C})}{\nabla}_\rho \overset{(2)}{\tilde{C}}{}^\rho_{\mu\nu} - 2\tilde{C}^\lambda_{[\mu\rho]} \overset{(2)}{\tilde{C}}{}^\rho_{\lambda\nu} - \overset{(2)}{\tilde{C}}{}^\rho_{\mu\lambda} S^\lambda_{\rho\nu}, \tag{2.25d}$$

$$\overset{(4)}{R}_{\mu\nu} = -\overset{(2)}{\tilde{C}}{}^\rho_{\mu\lambda} \overset{(2)}{\tilde{C}}{}^\lambda_{\rho\nu}. \tag{2.25e}$$

Note that the terms $S^\rho_{\mu\lambda} S^\lambda_{\rho\nu}$ and $S^\rho_{\rho\lambda} S^\lambda_{\mu\nu}$ at order $c^0$ vanish identically. Here, $\overset{(\tilde{C})}{\nabla}$ denotes the PUL covariant derivative and $\overset{(\tilde{C})}{R}_{\mu\nu}$ denotes its Ricci tensor. We can decompose the Levi-Civita Ricci scalar in a similar way,

$$R = \left( -\frac{1}{c^2} V^\mu V^\nu + \Pi^{\mu\nu} \right) R_{\mu\nu} \tag{2.26}$$

$$= -\frac{1}{c^4} V^\mu V^\nu \overset{(-2)}{R}_{\mu\nu} + \frac{1}{c^2} \left( -V^\mu V^\nu \overset{(0)}{R}_{\mu\nu} + \Pi^{\mu\nu} \overset{(-2)}{R}_{\mu\nu} \right)$$

$$- V^\mu V^\nu \overset{(2)}{R}_{\mu\nu} + \Pi^{\mu\nu} \overset{(0)}{R}_{\mu\nu} + c^2 \left( -V^\mu V^\nu \overset{(4)}{R}_{\mu\nu} + \Pi^{\mu\nu} \overset{(2)}{R}_{\mu\nu} \right). \tag{2.27}$$

Several of these terms simplify significantly once the expressions (2.25) are inserted. In particular, the $V^\mu V^\nu$ projection of the Ricci tensor $\overset{(\tilde{C})}{R}_{\mu\nu}$ associated to $\tilde{C}^\rho_{\mu\nu}$ vanishes due to its metric-compatibility, see Equation (A.12a).

## 2.5   Spatial hypersurfaces and projected connection

When constructing concrete solutions to the theories we will obtain in the following section, it will often be useful to specialize to a spatial hypersurface. However, as it stands, this concept is not well-defined in a Carroll manifold. The reason for this is that we can currently only define spatial covectors but not spatial vectors,

$$v^\mu X_\mu = 0, \tag{2.28}$$

since $\tau_\mu Y^\mu$ is generically not boost-invariant. For this reason, Carroll geometry naturally induces a fiber bundle structure on the spacetime manifold, where the worldlines of $v^\mu$ are the one-dimensional fibers, as emphasized in [39]. The horizontal sections of this fiber bundle then correspond to spatial submanifolds, which is equivalent to choosing an integrable Ehresmann connection. The freedom of choice in the Ehresmann corresponds to the transformation of $\tau_\mu$ under local Carroll boosts.

To define a spatial hypersurface, we must therefore specialize to a particular boost frame. This is in contrast to Newton–Cartan geometry, where the Galilei boosts leave $\tau_\mu$ invariant and the natural spacetime fibration is one of spatial hypersurfaces instead of time lines. As we will see, in Carroll geometry, we can frequently perform our spatial hypersurface computations in one such boost frame and subsequently extrapolate to frame-independent quantities on the entire spacetime.

For this, we choose coordinates $x^\mu = (t, x^i)$ such that the Carroll boost-invariant vector field $v^\mu$ is parallel to a coordinate vector, where $i = 1, \ldots, d$. It will be useful to allow for a 'lapse' function $\alpha$, so that we have

$$v^\mu \partial_\mu = \alpha^{-1} \partial_t. \tag{2.29}$$

In these coordinates the one-form $\tau_\mu$ can be decomposed as

$$\tau_\mu dx^\mu = -\alpha dt + b_i dx^i, \tag{2.30}$$

where $b_i$ corresponds to the Ehresmann connection in [39]. In this coordinate frame, the local Carroll boost transformations (2.9) act by shifting $b_i \to b_i + \lambda_i$, corresponding to a different choice of Ehresmann connection. Using the gauge freedom afforded by the boost transformations, we can therefore always at least locally go to an equivalent frame where $b_i = 0$. In this boost frame, the tangent subspace defined by the kernel of $\tau_\mu$ can be integrated to a spatial foliation given by constant $t$-slices. The integrability of the spatial slices is also reflected by the fact that its defining one-form $\tau_\mu$ then satisfies the Frobenius condition $\tau \wedge d\tau = 0$. Consequently, in this frame, the Carroll metric data is given by

$$v^\mu \partial_\mu = \alpha^{-1} \partial_t, \qquad \tau_\mu dx^\mu = -\alpha dt, \tag{2.31a}$$

$$h_{\mu\nu} dx^\mu dx^\nu = h_{ij} dx^i dx^j, \qquad h^{\mu\nu} \partial_\mu \partial_\nu = h^{ij} \partial_i \partial_j, \tag{2.31b}$$

so that $h_{ij}$ is a Riemannian metric on the spatial hypersurfaces, with $h^{ij}$ as its inverse.

Additionally, we can define the spatial and temporal projectors as

$$h^\mu_\nu = h^{\mu\rho} h_{\rho\nu}, \qquad -v^\mu \tau_\nu. \tag{2.32}$$

Using these, we can define a projected covariant derivative on the spatial slices,

$$\hat{\nabla}_\rho X_{\mu_1 \dots \mu_l}{}^{\nu_1 \dots \nu_k} = h^\gamma_\rho h^{\alpha_1}_{\mu_1} \dots h^{\alpha_l}_{\mu_l} h^{\nu_1}_{\beta_1} \dots h^{\nu_k}_{\beta_k} \tilde{\nabla}_\gamma X_{\alpha_1 \dots \alpha_l}{}^{\beta_1 \dots \beta_k}. \tag{2.33}$$

This projected covariant derivative acts on spatial tensors and can therefore be understood as an intrinsic operator on the spatial slices. A small calculation shows that it is compatible with the spatial projector,

$$\hat{\nabla}_\rho h^\mu_\nu = h^\mu_\alpha h^\beta_\nu h^\gamma_\rho \tilde{\nabla}_\gamma \left( h^{\alpha\lambda} h_{\lambda\beta} \right) = h^\mu_\alpha h_{\nu\lambda} h^\gamma_\beta \left( -v^{(\alpha} h^{\lambda)\sigma} \tau_{\sigma\kappa} \left[ \delta^\kappa_\gamma - v^\kappa \tau_\gamma \right] \right) = 0, \tag{2.34}$$

so that we can consistently contract spatial indices inside the derivative. Here, we have defined $\tau_{\mu\nu} = 2\partial_{[\mu} \tau_{\nu]}$ and we have used the fact that $h^{\mu\rho} h^{\nu\sigma} \tau_{\rho\sigma} = 0$, which follows from the Frobenius condition. Next, one can show that the projected covariant derivative is torsionless and compatible with the spatial metric and its inverse,

$$[\hat{\nabla}_\mu, \hat{\nabla}_\nu] f = 0, \qquad \hat{\nabla}_\rho h_{\mu\nu} = \hat{\nabla}_\rho h^{\mu\nu} = 0, \tag{2.35}$$

where $f$ is a scalar function. This implies that the projected connection is the Levi-Civita connection constructed from the spatial metric.

The equivalent of the Gauss equation then allows us to write its curvature in terms of the spacetime curvature,

$$\hat{R}_{\mu\nu\rho}{}^\sigma = h^\alpha_\mu h^\beta_\nu h^\gamma_\rho h^\sigma_\delta \tilde{R}_{\alpha\beta\gamma}{}^\delta. \tag{2.36}$$

Finally, one can show that the Ricci tensor of the projected connection is related to the spacetime Ricci tensor by

$$\hat{R}_{\mu\nu} = h^\alpha_\mu h^\beta_\nu \tilde{R}_{\alpha\beta} + \hat{\nabla}_\mu a_\nu + a_\mu a_\nu, \tag{2.37}$$

where $a_\mu = \mathcal{L}_v \tau_\mu = v^\nu \tau_{\nu\mu}$ is the acceleration one-form. Using these equations, the curvature tensors that enter in our actions for Carroll gravity below can be related to the induced curvature tensors on spatial slices. Although such spatial slices are not a boost-invariant notion in Carroll geometry, we will see in Section 4 below that the resulting equations can be a useful tool for constructing solutions from which we can subsequently reconstruct a boost-invariant Carroll geometry.

# 3 Carroll gravity from expanding general relativity

We now have all the ingredients needed to develop a systematic and covariant ultra-local expansion of general relativity. First, using the results from the previous section, we rewrite the Einstein–Hilbert action in terms of the pre-ultra-local (PUL) variables. Next, we can expand the resulting PUL action to a particular order in $c^2$ in terms of the leading-order Carroll variables and their subleading corrections. In the following, we mainly focus on the leading-order theory, but we also consider a subsector of the next-to-leading order theory. Next, we show that these theories are equivalent to the 'electric' and 'magnetic' limits discussed in [21, 47], providing for the first time a covariant description of the latter using our Lagrangian perspective. Finally, we note a useful analogy between the resulting equations of motion and the constraint and evolution equations appearing in the 3+1 decomposition of general relativity.

## 3.1 'Pre-ultra-local' Einstein–Hilbert action

Using the expression (2.27) for the Levi-Civita Ricci scalar, we can rewrite the Einstein–Hilbert action in terms of the PUL variables as follows,

$$S_{\text{EH}} = \frac{c^3}{16\pi G} \int_M R \sqrt{-g}\, d^{d+1}x \tag{3.1}$$

$$\approx \frac{c^2}{16\pi G} \int_M \left[ \left( \mathcal{K}^{\mu\nu}\mathcal{K}_{\mu\nu} - \mathcal{K}^2 \right) + c^2 \Pi^{\mu\nu}\overset{(c)}{R}_{\mu\nu} + \frac{c^4}{4} \Pi^{\mu\nu}\Pi^{\rho\sigma} (dT)_{\mu\rho} (dT)_{\nu\sigma} \right] E\, d^{d+1}x, \tag{3.2}$$

which holds up to boundary terms. The PUL parametrization (3.2) puts the Einstein–Hilbert action in a form that prepares it for the ultra-local expansion. Performing this expansion, we obtain a series of actions describing covariant theories of dynamical Carroll geometry plus subleading corrections,

$$S_{\text{EH}} = c^2 \overset{(2)}{S}_{\text{LO}} + c^4 \overset{(4)}{S}_{\text{NLO}} + \mathcal{O}(c^6). \tag{3.3}$$

At leading order (LO), we find a theory with local Carroll symmetry. Each subsequent step in the expansion adds additional fields and interpolates further between the Carroll-invariant theory at LO and the full relativistic Einstein–Hilbert theory.

## 3.2 Leading-order and next-to-leading-order theory

Using the expansion of the PUL variables in (2.4), we see that the LO term coming from the PUL Einstein–Hilbert action is given by

$$\overset{(2)}{S}_{\text{LO}} = \frac{1}{16\pi G} \int_M \left[ K^{\mu\nu}K_{\mu\nu} - K^2 \right] e\, d^{d+1}x, \tag{3.4}$$

where $e = \det(\tau_\mu, e_\mu{}^a)$ is the Carroll-invariant vielbein determinant. Note that this action depends only on the extrinsic curvature $K_{\mu\nu} = -\frac{1}{2}\mathcal{L}_\nu h_{\mu\nu}$ and not on the curvature $\tilde{R}^\rho{}_{\mu\nu\sigma}$ associated to the LO Carroll connection $\tilde{\Gamma}^\rho_{\mu\nu}$. This action was previously obtained from a 'zero signature' limit in the Hamiltonian formulation of GR [23], which has recently been revisited as an 'electric' Carroll limit of general relativity [47]. We will return to the corresponding 'magnetic' limit and its relation to our expansion below. Additionally, a class of actions including (3.4) have been obtained from gauging the Carroll algebra [27]. Here, we see that the action (3.4) is the natural starting point in an ultra-local expansion of general relativity.

To study its dynamics, we can vary the LO action (3.4) with respect to $v^\mu$ and $h^{\mu\nu}$,

$$\delta \overset{(2)}{S}_{\text{LO}} \approx \frac{1}{8\pi G} \int_M \left[ \overset{(2)}{G}{}^\nu_\mu \delta v^\mu + \frac{1}{2}\overset{(2)}{G}{}^h_{\mu\nu} \delta h^{\mu\nu} \right] e\, d^{d+1}x, \tag{3.5}$$

which leads to the LO equations of motion $\overset{(2)}{G}{}^\nu_\mu = 0$ and $\overset{(2)}{G}{}^h_{\mu\nu} = 0$. They are given by

$$\overset{(2)}{G}{}^\nu_\mu = -\frac{1}{2}\tau_\mu\left(K^{\rho\sigma}K_{\rho\sigma} - K^2\right) + h^{\nu\rho}\tilde\nabla_\rho\left(K_{\mu\nu} - Kh_{\mu\nu}\right), \tag{3.6a}$$

$$\overset{(2)}{G}{}^h_{\mu\nu} = -\frac{1}{2}h_{\mu\nu}\left(K^{\rho\sigma}K_{\rho\sigma} - K^2\right) + K\left(K_{\mu\nu} - Kh_{\mu\nu}\right) - v^\rho\tilde\nabla_\rho\left(K_{\mu\nu} - Kh_{\mu\nu}\right). \tag{3.6b}$$

Note that we have varied with respect to $h^{\mu\nu}$, which transforms under boosts, but the resulting Ward identity $v^\mu\overset{(2)}{G}{}^h_{\mu\nu} = 0$ guarantees that this is not a problem. Using the variables $(v^\mu, h^{\mu\nu})$ has the advantage that it fully specifies the inverse pair $(\tau_\mu, h_{\mu\nu})$ at the price of boost transformations, whereas the pair $(\tau_\mu, h^{\mu\nu})$ cannot be solved from $(v^\mu, h_{\mu\nu})$ since the latter two are boost-invariant. We have chosen the above since we want to keep boost transformations explicit in most of the following.

In contrast to the LO action in the non-relativistic expansion (see Equation (3.9) of [6]), which only serves to impose a foliation-preserving condition on the leading-order Newton–Cartan variables, this action already contains non-trivial physical solutions, which we will study in more detail in Section 4. Indeed, looking at the PUL parametrization (3.2) of the Einstein–Hilbert action, we see that these solutions probe the 'kinetic' terms of the relativistic Lagrangian at order $c^2$, rather than the 'foliation' terms at order $c^6$. One of the virtues of studying the ultra-local expansion is that it allows us to explore different parts of the full relativistic theory in a more controlled setting.

**Interpretation as constraint and evolution equations**

Before proceeding to the NLO theory, it is instructive to rewrite the LO equations (3.6) using the natural split between time and space that appears in the Carroll geometry. Projecting out the time and space components of each equation using $v^\mu$ and $h^{\mu\nu}$, we see that the time component of $\overset{(2)}{G}{}^h_{\mu\nu}$ vanishes. The remaining three sets of equations can then be written as

$$K^{\mu\nu}K_{\mu\nu} - K^2 = 0, \tag{3.7a}$$

$$h^{\rho\sigma}\tilde\nabla_\rho(K_{\sigma\mu} - Kh_{\sigma\mu}) = 0, \tag{3.7b}$$

$$\mathcal{L}_v K_{\mu\nu} = -2K_\mu{}^\rho K_{\rho\nu} + KK_{\mu\nu}. \tag{3.7c}$$

To get to the third equation, we used (3.7a) and exchanged the time derivative $v^\rho\tilde\nabla_\rho$ for a Lie derivative using the formula (A.6) applied to our connection (2.14).

In the form (3.7), the LO equations of motion are strongly reminiscent of the 3+1 decomposition of the Einstein equation. The first two equations (3.7a) and (3.7b) can be interpreted as *constraint* equations, meaning that they can be checked on a single 'equal-time slice' in the sense of Section 2.5, and they restrict what kind of initial data $(h_{\mu\nu}, K_{\mu\nu})$ on such a slice is valid. Given such initial data, Equation (3.7c) and the definition $K_{\mu\nu} = -\frac{1}{2}\mathcal{L}_v h_{\mu\nu}$ of the extrinsic curvature then dictate its evolution along the 'time' direction defined by the vector field $v^\mu$.

Remarkably, in contrast to the 3+1 decomposition of the Einstein equations, the right-hand side of (3.7c) contains no spatial derivatives. Thus, starting from a point on an equal-time slice, the evolution of the initial data along $v^\mu$ depends only on the value of the extrinsic curvature and spatial metric at that particular point. This reflects the intuition that, in the ultra-local Carroll limit, the light cone collapses to a line, and causal evolution proceeds independently on distinct integral lines of $v^\mu$. As a result, using adapted coordinates, the initial-value problem is simplified from a highly non-trivial PDE in general relativity to an integrable ODE. (A similar observation was made by Dautcourt in [24].) We discuss this in more detail in Section 4.1, where we analytically solve for the time evolution of arbitrary initial data. Additionally, in

$$\overset{(2)}{G}{}^{h}_{\mu\nu} \qquad\qquad \overset{(2)}{G}{}^{v}_{\mu}$$

$$// \qquad\qquad\qquad\qquad \backslash\backslash$$

$$\overset{(4)}{G}{}^{\Phi}_{\mu\nu} \qquad \overset{(4)}{G}{}^{h}_{\mu\nu} \qquad \overset{(4)}{G}{}^{v}_{\mu} \qquad \overset{(4)}{G}{}^{M}_{\mu}$$

Figure 1: The ultra-local $c^2$ expansion of the Einstein–Hilbert action leads to two new equations of motion for each order in the expansion. Every order also contains the equations of the preceding order, which is illustrated up to NLO above [6, 78, 79].

Section 4.3, we adapt well-known methods for constructing relativistic initial data from the 3+1 formalism to construct non-trivial solutions for our LO Carroll theory.

**Next-to-leading-order theory**

Following the general procedure developed for the non-relativistic Galilei $1/c^2$ expansion in [6, 78], the NLO action in the ultra-local Carroll $c^2$ expansion (3.3) of GR is

$$\overset{(4)}{S}_{\text{NLO}} = \frac{1}{16\pi G} \int_M \left[ h^{\mu\nu}\tilde{R}_{\mu\nu} + 2\overset{(2)}{G}{}^{v}_{\mu}M^{\mu} + \frac{1}{2}\overset{(2)}{G}{}^{h}_{\mu\nu}\Phi^{\mu\nu} \right] e\, d^{d+1}x. \tag{3.8}$$

Here, $M^{\mu}$ and $\Phi^{\mu\nu}$ are the NLO variables appearing in the expansion (2.4) of the PUL variables $V^{\mu}$ and $\Pi^{\mu\nu}$. The equations of these subleading variables are equal to the equations of motion (3.6) of the corresponding LO variables $v^{\mu}$ and $h^{\mu\nu}$ in the LO action (3.4). For this reason, the NLO action (3.8) contains all dynamics associated to the LO action. This is illustrated in Figure 1.

In addition, the NLO action leads to a more complicated set of equations of motion for the LO variables $v^{\mu}$ and $h^{\mu\nu}$. In particular, these equations now involve variations of the Ricci curvature $\tilde{R}_{\mu\nu}$ that was missing from the LO action (3.4), which allow the theory to support massive solutions, as we will show in Section 4.4. For this, we only need a subsector of the NLO equations of motion, and we leave the study of the full NLO theory to future work.

### 3.3 Truncation and relation to 'magnetic' Carroll gravity

As a first step in studying the dynamics of the NLO action (3.8), we can consider a truncation of the theory where the NLO fields $M^{\mu}$ and $\Phi^{\mu\nu}$ are set to zero by hand:

$$\overset{(4)}{S}_{\text{NLO}}\Big|_{M^{\mu}=\Phi^{\mu\nu}=0} = \frac{1}{16\pi G} \int_M h^{\mu\nu}\tilde{R}_{\mu\nu}\, e\, d^{d+1}x. \tag{3.9}$$

This truncation no longer reproduces the equations of motion of the LO theory. However, the resulting equations of motion for the remaining variables $v^{\mu}$ and $h^{\mu\nu}$ are significantly simpler than those of the full NLO action. Up to boundary terms, we get

$$\delta\overset{(4)}{S}_{\text{NLO}}\Big|_{M^{\mu}=\Phi^{\mu\nu}=0} \approx \frac{1}{8\pi G} \int_M \left[ \overset{(4)}{G}{}^{v}_{\mu}\delta v^{\mu} + \frac{1}{2}\overset{(4)}{G}{}^{h}_{\mu\nu}\delta h^{\mu\nu} \right]\Big|_{M^{\mu}=\Phi^{\mu\nu}=0} e\, d^{d+1}x, \tag{3.10}$$

where the truncated NLO equations of motion are given by

$$\overset{(4)}{G}{}_{\mu}^{v}\Big|_{M^{\mu}=\Phi^{\mu v}=0} = \tau_{\mu}h^{\rho\sigma}\tilde{R}_{\rho\sigma} + \frac{1}{2}h^{v\sigma}h_{\mu}^{\lambda}\tau_{v\lambda}v^{\kappa}\tau_{\kappa\sigma} - \tilde{\nabla}_{v}\left(h^{v\sigma}\tau_{\mu\sigma}\right), \tag{3.11a}$$

$$\overset{(4)}{G}{}_{\mu v}^{h}\Big|_{M^{\mu}=\Phi^{\mu v}=0} = \tilde{R}_{(\mu v)} - \frac{1}{2}h_{\mu v}h^{\sigma\rho}\tilde{R}_{\sigma\rho} + 2Kv^{\kappa}\tau_{\kappa(\mu}\tau_{v)} + \tau_{\sigma\rho}K_{(\mu}{}^{\sigma}\left(v^{\rho}\tau_{v)} - \delta_{v)}^{\rho}\right) \tag{3.11b}$$

$$- \tilde{\nabla}_{\lambda}\left(Kh_{(\mu}^{\lambda}\tau_{v)} - K^{\lambda}{}_{(\mu}\tau_{v)} - \frac{1}{2}h_{\mu v}h^{\lambda\gamma}v^{\kappa}\tau_{\kappa\gamma}\right).$$

Note that solutions of these equations are only solutions of the full NLO equations if they also satisfy the LO equations of motion, which are not contained in the truncated action (3.9). As we show in Section 4.4, the presence of the curvature terms in (3.11) allows us to obtain massive solutions. Such terms are absent from the LO theory and first appear at NLO. Next, we consider a further simplification of the NLO theory.

**Relation to 'magnetic' Carroll limit of general relativity**

In recent work, 'electric' and 'magnetic' Carroll limits of field theories and gravity were studied from a Hamiltonian perspective in [47] and also for field theories from a Lagrangian perspective in [21]. This terminology is motivated by the two distinct Carroll limits of electromagnetism that retain the electric and magnetic sectors, respectively [26]. We can now complete this picture by proposing a Lagrangian approach to both the electric and the magnetic limit of limits gravity in terms of our general $c^2$ expansion, following the suggestion of [21].

As we have already remarked above, our LO theory (3.4) agrees with the electric limit of gravity that was obtained in [47] and also earlier in [23]. This theory describes the dominant dynamics of gravity in the $c \to 0$ ultra-local Carroll limit.

Next, to connect to the magnetic limit, we rewrite the PUL form of the Einstein–Hilbert action (3.2) as follows,

$$S = \frac{c^4}{16\pi G}\int_{M}\left[-\frac{c^2}{4}G^{\mu v\rho\sigma}\chi_{\mu v}\chi_{\rho\sigma} + G^{\mu v\rho\sigma}\chi_{\mu v}\mathcal{K}_{\rho\sigma}\right. \tag{3.12}$$

$$\left. + \Pi^{\mu v}\overset{(C)}{\tilde{R}}_{\mu v} + \frac{c^2}{4}\Pi^{\mu v}\Pi^{\rho\sigma}(dT)_{\mu\rho}(dT)_{v\sigma}\right]E\,d^{d+1}x.$$

Here, we have introduced an auxiliary symmetric spatial tensor $\chi_{\mu v}$, i.e. $V^{\mu}\chi_{\mu v} = 0$. In addition, we introduced the DeWitt metric and its inverse,

$$G^{\mu v\rho\sigma} = \frac{1}{2}\left(\Pi^{\mu\rho}\Pi^{v\sigma} + \Pi^{\mu\sigma}\Pi^{v\rho} - 2\Pi^{\mu v}\Pi^{\rho\sigma}\right), \tag{3.13a}$$

$$G_{\mu v\rho\sigma} = \frac{1}{2}\left(\Pi_{\mu\rho}\Pi_{v\sigma} + \Pi_{\mu\sigma}\Pi_{v\rho} - \frac{2}{d-1}\Pi_{\mu v}\Pi_{\rho\sigma}\right). \tag{3.13b}$$

Note that these tensors satisfy $G_{\mu v\kappa\lambda}G^{\kappa\lambda\rho\sigma} = \Pi_{(\mu}^{\rho}\Pi_{v)}^{\sigma}$, where $\Pi_{v}^{\mu} = \delta_{v}^{\mu} + V^{\mu}T_{v}$. Integrating out $\chi_{\mu v}$, we see that this action is equivalent to the PUL Einstein–Hilbert action (3.2). On the other hand, taking $c \to 0$ in (3.12) leads to the action

$$\overset{(4)}{S}_{\text{mag}} = \frac{1}{16\pi G}\int_{M}\left[\phi^{\mu v}K_{\mu v} + h^{\mu v}\tilde{R}_{\mu v}\right]e\,d^{d+1}x, \tag{3.14}$$

where we have introduced $\phi^{\mu v} = G^{\mu v\rho\sigma}\chi_{\rho\sigma}$. This action is invariant under Carroll boosts provided $\phi^{\mu v}$ transforms as

$$\delta_{\lambda}\phi^{\mu v} = 2\left[-h^{\mu v}\lambda^{\rho}\tau_{\rho\sigma}v^{\sigma} + h^{\rho(\mu}\lambda^{v)}\tau_{\rho\sigma}v^{\sigma} - h^{\rho(\mu}\tilde{\nabla}_{\rho}\lambda^{v)} + h^{\mu v}h^{\rho}{}_{\sigma}\tilde{\nabla}_{\rho}\lambda^{\sigma}\right]. \tag{3.15}$$

Note that a similar Carroll-invariant theory of gravity was previously introduced from a first-order perspective in [30]. As we will now show, the action (3.14) is equivalent to the magnetic limit of general relativity proposed in [47]. On the one hand, the field $\phi^{\mu\nu}$ plays the role of a Lagrange multiplier imposing the constraint $K_{\mu\nu} = 0$. On the other hand, in a Hamiltonian analysis, it can be interpreted as the momentum dual to $h_{\mu\nu}$, since only the first term of our magnetic action contains time derivatives. The remaining equations of motion, coming from $v^\mu$ and $h^{\mu\nu}$, are given by

$$0 = \tilde{\nabla}_\rho \phi^\rho{}_\mu - v^\rho \tau_{\rho\nu} \phi^\nu{}_\mu + \tau_\mu h^{\rho\sigma} \tilde{R}_{\rho\sigma} + \frac{1}{2} h^{\nu\sigma} h^\lambda_\mu \tau_{\nu\lambda} v^\kappa \tau_{\kappa\sigma} - \tilde{\nabla}_\nu \left( h^{\nu\sigma} \tau_{\mu\sigma} \right), \qquad (3.16a)$$

$$0 = -\frac{1}{2} v^\rho \tilde{\nabla}_\rho \phi_{\mu\nu} + \tilde{R}_{\mu\nu} - \frac{1}{2} h_{\mu\nu} h^{\sigma\rho} \tilde{R}_{\sigma\rho} + \frac{1}{2} h_{\mu\nu} \tilde{\nabla}_\lambda (h^{\lambda\gamma} v^\kappa \tau_{\kappa\gamma}). \qquad (3.16b)$$

Note that the Ricci tensor $\tilde{R}_{\mu\nu}$ is symmetric once we impose the constraint $K_{\mu\nu}$, see Equation (A.14). Later on, we will see that these equations allow for solutions with a non-zero mass, in contrast to the leading-order electric theory. A similar observation was recently made in an asymptotic symmetry analysis of the magnetic theory [48].

Next, following the discussion in Section 2.5, we can go to a boost frame that allows for a spatial foliation. In terms of the curvature of the projected derivative (2.33), the spatial and temporal projections of the equations of motion (3.16) are then given by

$$0 = h^{\mu\nu} \hat{R}_{\mu\nu}, \qquad (3.17a)$$

$$0 = \hat{\nabla}_\nu \phi^\nu{}_\mu, \qquad (3.17b)$$

$$\frac{1}{2} \mathcal{L}_\nu \phi_{\mu\nu} = \hat{R}_{\mu\nu} - \hat{\nabla}_{(\mu} a_{\nu)} - a_\mu a_\nu + h_{\mu\nu} h^{\rho\sigma} \left( \hat{\nabla}_{(\rho} a_{\sigma)} + a_\rho a_\sigma \right). \qquad (3.17c)$$

The temporal projection of (3.16b) vanishes. In addition, $K_{\mu\nu} = -\frac{1}{2} \mathcal{L}_\nu h_{\mu\nu} = 0$ must hold on all slices. Equation (3.17c) is an evolution equation for $\phi_{\mu\nu}$, whereas the first two equations precisely reproduce the two constraint of the Hamiltonian definition of the magnetic theory that was put forward in [47]. Therefore, following similar constructions in field theory [21], we can now identify the magnetic limit of general relativity as a truncated sector of the NLO theory in the ultra-local Carroll expansion.

# 4 Solutions of LO and NLO theories

Having constructed the LO theory and a subsector of the NLO theory in the ultra-local Carroll expansion of general relativity, which we subsequently identified with the electric and magnetic contractions, we now study solutions of these theories.

As we saw previously, we can split the equations of motion of the LO theory into constraint and evolution equations. First, starting from arbitrary initial data, we show that we can solve the evolution *analytically*, since it reduces to an ODE in suitable coordinates. Next, we adapt methods from the 3+1 formulation of general relativity and construct a Carroll version of the Bowen–York initial data. In general relativity, this describes black hole solutions of the constraint equations that are parametrized by their mass and their angular and linear momentum. We construct the analogue of these solutions in the LO Carroll theory and subsequently show that their momentum parameters correspond to conserved boundary charges.

Finally, we construct solutions with non-zero mass, corresponding to a Carroll limit of the Schwarzschild metric. Such solutions are absent in the LO theory, as was remarked in [48], but they can be obtained in the magnetic truncation of the NLO theory. We also incorporate a cosmological constant in the LO and magnetic theory.

### 4.1 Exact evolution of general initial data at LO

We now rewrite the LO evolution equation using the adapted coordinates that were introduced in Section 2.5. For this, it is convenient to reparametrize the lapse function as $\alpha = e^{-\frac{B}{2}}$. In these coordinates, the extrinsic curvature and its Lie derivative along $v^\mu$ then take the following form,

$$K_{ij} = -\frac{1}{2}\mathcal{L}_v h_{ij} = -\frac{e^{-\frac{B}{2}}}{2}\dot{h}_{ij}, \tag{4.1a}$$

$$\mathcal{L}_v K_{ij} = -\frac{e^{-B}}{2}\left(\ddot{h}_{ij} - \frac{\dot{B}}{2}\dot{h}_{ij}\right), \tag{4.1b}$$

where dots denotes differentiation with respect to the $t$ coordinate. The LO evolution equation (3.7c) then corresponds to the following ODE,

$$\ddot{h}_{ij} + \frac{1}{2}\dot{h}_{ij}(h^{kl}\dot{h}_{kl} - \dot{B}) - \dot{h}_{ik}h^{kl}\dot{h}_{kj} = 0. \tag{4.2}$$

We can simplify this equation by setting $\dot{B} = h^{ij}\dot{h}_{ij}$, which fixes the gauge freedom in the lapse function up to an overall shift. Finally, given the initial data $h_{ij}(t=0) = h_{(0)ij}$ and $K_{ij}(t=0) = K_{(0)ij}$, the solution to the evolution equation (4.2) is given by

$$h_{ij}(t) = h_{(0)ik}\exp[-2t\,h_{(0)}^{kl}K_{(0)lj}]. \tag{4.3}$$

In this form, the fact that $h_{ij}(t)$ is symmetric is perhaps not obvious, but it follows easily from the matrix identity $Ae^{A^{-1}B} = e^{BA^{-1}}A$ for square matrices $A, B$.

Remarkably, we see that the time evolution of arbitrary initial data can be solved analytically in the LO theory. This corresponds to the intuition that, in the ultra-local Carroll limit, the domain of dependence shrinks to a line, which trivializes the possible time-dependence of the theory. Similar results have been obtained in the context of the 'strong coupling' limit of general relativity [80]. Furthermore, this result (as well as the solutions obtained in Section 4.3 below) is strongly reminiscent of the Belinski-Khalatnikov-Lifshitz limit of general relativity [25]. It would be very interesting to further investigate these relations and to see how subleading corrections modify the possible dynamics that can arise.

### 4.2 Conserved boundary charges

An important way to characterize solutions in general relativity is by computing their conserved boundary charges. We can develop a corresponding notion of boundary charges for the Carroll LO theory governed by the action (3.4). A related asymptotic symmetry analysis was done for the equivalent electric theory in the Hamiltonian formalism in [48]. Killing vector fields $\xi^\mu$ of the LO Carroll geometry can be defined as

$$\mathcal{L}_\xi v^\mu = 0, \qquad \mathcal{L}_\xi h_{\mu\nu} = 0. \tag{4.4}$$

The boundary charge associated to such a Carroll Killing vector field $\xi$ can be computed using the covariant phase space formalism [81, 82] (see also [83]). This formalism allows one to compute a charge integrand $k_\xi^{[\mu\nu]}$ corresponding to $\xi$ and a variation of the metric data. This charge integrand can be computed using

$$k_\xi^{[\mu\nu]} = -\delta_h Q^{\mu\nu} + 2\xi^{[\mu}\Theta_h^{\nu]}, \tag{4.5}$$

where $\delta$ indicates an on-shell variation of the metric data, $Q^{\mu\nu}$ is the Noether-Wald charge and $\Theta^\mu$ is the presymplectic potential. For the LO theory, these are given by

$$\Theta^\mu = \frac{e}{8\pi G}\left[(Kh^\mu_{\ \nu} - K^\mu_{\ \nu})\delta v^\nu - \frac{1}{2}(Kh_{\sigma\rho} - K_{\sigma\rho})v^\mu \delta h^{\sigma\rho}\right], \tag{4.6}$$

$$Q^{[\mu\nu]} = \frac{e}{4\pi G}(v^{[\mu}K^{\nu]}_{\ \ \sigma}\xi^\sigma - v^{[\mu}\xi^{\nu]}K). \tag{4.7}$$

Integrated over a codimension two surface $S$ at infinity, it computes the change in the conserved charge associated to $\xi$ under the variation of the metric data,

$$\oint H_\xi = \int_S k^{\mu\nu}(d^{d-1}x)_{\mu\nu}. \tag{4.8}$$

The notation $\oint$ indicates that the charge is not necessarily integrable. If it is, we can integrate along a path in phase space from a reference metric to obtain a finite charge.

In the LO theory, it follows from the charge integrand (4.5) that there can be no non-zero charge associated to Killing vectors that are proportional to the Carroll vector $v^\mu$, which generates time translations. As a result, there seems to be no notion of mass or energy at the LO level of the ultra-local expansion of general relativity.

## 4.3 Bowen–York-type solutions

In the previous section we saw that any initial data can be analytically evolved forward in time. Hence, we can easily get a complete solution of the LO equations of motion after solving the constraint equations (3.7a)-(3.7b) to obtain allowed initial data. In the 3+1 formulation of general relativity, creating initial data is a well-studied subject. Due to the similarities of the LO constraint equations and the full relativistic constraints, the relativistic methods can be adapted to the Carroll limit, as we will now demonstrate.

As a first non-trivial set of solutions of the LO constraint equations, we consider a set of initial data that is similar to the so-called Bowen–York solutions [74] (see also [84]). In general relativity, this initial data corresponds to a black hole, which is parameterized by its mass and its linear and angular momenta $P_i$ and $J_i$. As we will see, only the momenta are retained in the LO solution we construct.

First, we start from a conformally flat ansatz for the spatial metric data. In general relativity, the conformal factor is ultimately solved from a non-linear Poisson equation to obtain the full Bowen–York initial data, which generically has to be done numerically. In contrast, the simplified form of the constraints that arises from the LO theory can be solved analytically. Specializing to $d = 3$, the Carroll LO Bowen–York-type initial data depends on the 7 parameters $(K_{(0)}, P_i, J_i)$ and can be written as

$$h_{(0)ij} = \psi^4 \delta_{ij}, \tag{4.9a}$$

$$K_{(0)ij} = \psi^{-2}\bar{L}X_{ij} + \frac{1}{3}K_{(0)}\psi^4 \delta_{ij}, \tag{4.9b}$$

where the conformal factor $\psi$ and the conformal Killing derivative $\bar{L}X^{ij}$ are given by

$$\psi = \left[\frac{3}{2(K_{(0)})^2}\bar{L}X_{ij}\bar{L}X^{ij}\right]^{1/12}, \tag{4.10a}$$

$$\bar{L}X^{ij} = \frac{3}{2r^3}\left[x^i P^j + x^j P^i - \left(\delta^{ij} - \frac{x^i x^j}{r^2}\right)P_k x^k\right] \tag{4.10b}$$

$$+ \frac{3}{r^5}\left[\epsilon^{ik}_{\ \ l}J_k x^l x^j + \epsilon^{jk}_{\ \ l}J_k x^l x^i\right].$$

Here, indices are raised and lowered using $\delta_{ij}$, and $\epsilon_{ijk}$ is the totally anti-symmetric symbol in 3 dimensions defined by $\epsilon_{123} = 1$. These solutions can be analytically evolved in time using the general prescription (4.3). We can check that the parameters $P_i$ and $J_i$ are equal to the charges associated to (asymptotic) translation and rotation symmetry using the boundary charges (4.8).

## 4.4 Massive Schwarzschild solutions at NLO

In Section 4.2, we found that there appear to be no massive solutions at LO in the ultra-local expansion of general relativity. This is rectified at NLO, as we will now show using the subsector corresponding to the magnetic theory (3.14). For this, we focus on static solutions with $\phi^{\mu\nu} = 0$ (and recall that $K_{\mu\nu} = 0$ in the magnetic theory). Following the discussion in Section 2.5, we can go to an equal-time slice in a boost frame where the Frobenius condition $\tau \wedge d\tau = 0$ holds. Using the projected covariant derivative $\hat{\nabla}$ and its curvature from (2.33) and (2.36), the NLO constraint and evolution equations (3.17) can then be written as

$$h^{\mu\nu}\hat{R}_{\mu\nu} = 0 \, , \tag{4.11a}$$

$$\hat{R}_{\mu\nu} = \hat{\nabla}_\mu a_\nu + a_\mu a_\nu \, , \tag{4.11b}$$

where $a_\mu = \mathcal{L}_\nu \tau_\mu$ is the acceleration one-form. Sxince there is no evolution, we are left with constraint equations only. In terms of the parametrization (2.31) with time and space coordinates $x^\mu = (t, x^i)$, we can relate this acceleration to the lapse $\alpha$ using

$$a_\mu = \alpha^{-1}\hat{\nabla}_\mu \alpha \, , \qquad a_\mu a_\nu + \hat{\nabla}_\mu a_\nu = \alpha^{-1}\hat{\nabla}_\mu \hat{\nabla}_\nu \alpha \, . \tag{4.12}$$

As a result, the constraints (4.11) can be written as

$$\hat{\nabla}^2 \alpha = 0 \, , \tag{4.13a}$$

$$\hat{R}_{ij} = \alpha^{-1}\hat{\nabla}_i \hat{\nabla}_j \alpha \, . \tag{4.13b}$$

To obtain a massive solution, we can specialize to $d = 3$ and consider a conformally flat metric $h_{ij} = \psi^4 \delta_{ij}$. Then (4.13a) implies $\partial^2 \psi = 0$, where $\partial^2$ is the flat space Laplacian. From this, we can obtain the solution

$$\psi = 1 + \frac{M}{2r} \, , \qquad \alpha = \frac{M - 2r}{M + 2r} \, , \tag{4.14}$$

which results in the following static solution of the NLO theory,

$$v^\mu \partial_\mu = \frac{M + 2r}{M - 2r}\partial_t \, , \qquad h_{\mu\nu}dx^\mu dx^\nu = \left(1 + \frac{M}{2r}\right)^4 \delta_{ij}dx^i dx^j \, . \tag{4.15}$$

This corresponds to the Carroll limit of the Schwarzschild solution in isotropic coordinates. Using the spatial curvature terms in (4.11), we see that we can construct massive solutions at NLO, in accordance with the observations for the magnetic theory in [48].

## 4.5 Cosmological constant solutions

Finally, we can add a cosmological constant $\Lambda$ to the PUL Einstein–Hilbert action (3.2),

$$S_\Lambda = \frac{c^4}{16\pi G}\int d^{d+1}x \, E(-2\Lambda) \, . \tag{4.16}$$

By considering different scalings of $\Lambda$ in $c^2$, we can introduce a cosmological constant term at different levels in the expansion. For example, if we consider

$$\Lambda = \frac{1}{c^2}\overset{(-2)}{\Lambda} + \overset{(0)}{\Lambda} + \cdots, \tag{4.17}$$

we see that $\overset{(-2)}{\Lambda}$ appears in the LO theory, while $\overset{(0)}{\Lambda}$ appears in the NLO theory, and so on. With these scalings, we will see that the LO theory requires $\overset{(-2)}{\Lambda}$ to be non-negative, while in the magnetic theory $\overset{(0)}{\Lambda}$ can be anything.

**Positive cosmological constant solutions of the LO theory**

The LO equations of motion (3.7) are modified by the cosmological constant as follows,

$$K^{\mu\nu}K_{\mu\nu} - K^2 = -2\overset{(-2)}{\Lambda}, \tag{4.18a}$$

$$h^{\rho\sigma}\tilde{\nabla}_\rho(K_{\sigma\mu} - Kh_{\sigma\mu}) = 0, \tag{4.18b}$$

$$\mathcal{L}_\nu K_{\mu\nu} = -2K_\mu{}^\rho K_{\rho\nu} + KK_{\mu\nu} - \frac{2}{d-1}\overset{(-2)}{\Lambda}h_{\mu\nu}. \tag{4.18c}$$

Following our discussion in Section 4.1, the evolution equation (4.18c) also admits a general analytic solution. Here, we focus on constructing a particular class of solutions characterized by uniform expansion. Using the adapted time and space coordinates $x^\mu = (t, x^i)$ from (2.31), we can consider the initial data

$$h_{(0)ij}, \qquad K_{(0)ij} = -Hh_{(0)ij}, \tag{4.19}$$

where $h_{(0)ij}$ is an arbitrary $d$-dimensional Riemannian metric and $H$ is a constant. For this class of initial data, the constraint (4.18b) is satisfied automatically due to metric-compatibility. The constraint (4.18a) implies that $\overset{(-2)}{\Lambda} = d(d-1)H^2/2$, which is always positive. Finally, with $\nu = \partial_t$, we find that the evolution equation (4.18c) gives

$$h_{\mu\nu}(t) = e^{2Ht}h_{(0)\mu\nu}, \qquad K_{\mu\nu}(t) = -Hh_{\mu\nu}(t). \tag{4.20}$$

Remarkably, we see that we can find a class of solutions with arbitrary initial metric data $h_{(0)\mu\nu}$ in the presence of a positive cosmological constant at LO. Clearly, not all of these solutions to the vacuum LO theory can descend from a Carroll limit of vacuum solutions to the Einstein equation. For the solution (4.20) to be interpreted in the context of an ultra-local expansion, one would have to add appropriate matter at subleading orders, and the resulting energy-momentum tensor would have to satisfy the corresponding energy conditions.

As a simple physical example, we can choose the initial spatial metric $h_{(0)ij}$ to be the standard flat metric $\delta_{ij}$ in Cartesian coordinates. The LO solution (4.20) can then be interpreted as a Carroll limit of the de Sitter metric in planar coordinates

$$ds^2 = -c^2 dt^2 + e^{2Ht}\delta_{ij}dx^i dx^j, \tag{4.21}$$

with the cosmological constant $\Lambda = d(d-1)H^2/2c^2$. For this reason, the positive cosmological constant solution (4.20) of the LO theory is a natural starting point for exploring the ultra-local expansion of general relativity in the context of cosmology [21].

**Cosmological constant solutions of the NLO theory**

In the presence of a cosmological constant (4.17), the equations of motion (4.11) of the magnetic subsector of the NLO theory are modified as follows,

$$h^{\mu\nu}(\hat{\nabla}_\mu a_\nu + a_\mu a_\nu) = -\frac{2\overset{(0)}{\Lambda}}{d-1}\,, \tag{4.22a}$$

$$\hat{R}_{\mu\nu} - \hat{\nabla}_\mu a_\nu - a_\mu a_\nu = \frac{2\overset{(0)}{\Lambda}}{d-1}h_{\mu\nu}\,. \tag{4.22b}$$

Unlike the LO theory, the magnetic theory can have both positive and negative cosmological constant. To see this, we consider the (anti-)de Sitter metric in static coordinates,

$$ds^2 = -c^2(1-kr^2)dt^2 + (1-kr^2)^{-1}dr^2 + r^2 d\Omega_{d-1}^2\,, \tag{4.23}$$

where $k = 2\Lambda/d(d-1)$ and $d\Omega_{d-1}^2$ is the round metric on the $(d-1)$-sphere. First, note that the extrinsic curvature of equal $t$ surfaces vanishes, so that we can easily construct a magnetic limit of the metric in these coordinates. In terms of the adapted coordinates defined in (2.31), the corresponding Carroll data is parametrized by

$$\alpha = \sqrt{1-kr^2}\,, \qquad h_{ij}dx^i dx^j = (1-kr^2)^{-1}dr^2 + r^2 d\Omega_{d-1}^2\,, \tag{4.24}$$

which can be checked to satisfy the equations of motion (4.22).

# 5 Discussion and outlook

In this paper, we have initiated the ultra-local expansion of general relativity (GR). In particular, we obtained the leading-order (LO) and next-to-leading-order NLO actions in terms of Carroll geometry and its subleading corrections. We have analyzed the equations of motion of the LO theory, corresponding to the 'electric' Carroll limit of GR, and demonstrated that it has non-trivial solutions, for which we computed the conserved boundary charges. Furthermore, we have obtained the equations of motion for a subsector of the NLO theory, which we subsequently identified with the magnetic Carroll limit of GR. In the latter theory, we also constructed non-trivial solutions related to black holes and (anti-)de Sitter spacetimes.

We conclude by mentioning a number of further directions. First, it would be interesting to work out the relation between our pre-ultra-local parametrization (as well as the earlier pre-non-relativistic parametrization) to the 3+1 decomposition of GR in more detail. In particular, it seems likely that similar methods to the ones we have adapted can be used both to simplify the non-relativistic expansion and to construct further non-trivial solutions. While we have obtained the full NLO action, we have only written down its equations of motion for the truncated case where the additional NLO fields are set to zero, which includes the aforementioned magnetic theory as a subsector. We leave the computation of the NLO equations of motion and a study of its evolution equation as well as more general solutions to future work. These techniques may also help to clarify the precise connection between the Galilean Newton–Cartan expansion of GR and the post-Newtonian expansion, which is a subject of ongoing work.

Next, as we briefly mentioned earlier, there appears to be a close connection to work on the strong coupling or gradient expansion of general relativity. The leading-order notion of geometry that arises in this approach was related to Carroll symmetry in [42]. Additionally, the exact evolution and the solutions to the constraint equations of our LO theory appear to be similar to results that have been obtained in [80]. Finally, there also appears to be a strong connection between our results for the LO theory and the Belinski-Khalatnikov-Lifshitz (BKL)

near-singularity limit of general relativity [25]. It would be very interesting to explore these connections further. Using our systematic covariant Carroll expansion it would for example be feasible to explore subleading corrections to the BKL limit coming from the full NLO theory.

Another important direction will be to examine the coupling of matter to the Carroll expansion of GR, mimicking the corresponding study for non-relativistic gravity in [77], by analyzing the small $c$ expansion of matter actions on general backgrounds. One can then apply this expansion to specific cases, such as probe particles and different types of matter. It would be interesting to use our results to study cosmological perturbations, following the relation between Carroll symmetry and cosmology identified in [21].

Carroll geometry is also relevant to flat-space holography, as well the membrane paradigm of black hole horizons. One could wonder whether the Carroll expansion examined in this work can provide further insights into these connections. Finally, in view of the relevance of different types of Newton–Cartan geometries in non-relativistic string theory (see for example Refs. [15–17, 85]), a further investigation into how strings probe a target spacetime Carroll geometry, and more generally the geometry obtained from Carroll expansions, could teach us more about novel corners of string theory.[3]

## Acknowledgements

We thank Jan de Boer, Watse Sybesma, Stefan Vandoren and Manus Visser for useful discussions. We especially thank Jelle Hartong for early collaboration on this project and many useful discussions. DH is supported by the Swiss National Science Foundation through the NCCR SwissMAP. NO and GO are supported in part by the project "Towards a deeper understanding of black holes with non-relativistic holography" of the Independent Research Fund Denmark (grant number DFF-6108-00340) and by the Villum Foundation Experiment project 00023086.

# A  Conventions and useful identities

Our spacetime dimension is $d + 1$ and we use the following sets of indices:

- $\mu, \nu, \rho, \dots$ for spacetime coordinate indices, $\mu = 0, 1, \dots, d$.

- $i, j, k, \dots$ for spatial coordinate indices, $i = 1, \dots, d$.

- $A, B, C, \dots$ for spacetime frame indices, $A = 0, 1, \dots d$.

- $a, b, c, \dots$ for spatial frame indices, $a = 1, \dots d$.

## A.1  Curvatures of general connections

For a general covariant derivative $\nabla_\mu$ with connection coefficients $\Gamma^\rho_{\mu\nu}$, we define the corresponding Riemann curvature tensor $R_{\mu\nu\rho}{}^\sigma$ and torsion tensor $T^\rho{}_{\mu\nu}$ through its action on a generic vector $X^\mu$ and a co-vector $\omega_\mu$,

$$[\nabla_\mu, \nabla_\nu] X^\rho = -R_{\mu\nu\sigma}{}^\rho X^\sigma - T^\sigma_{\mu\nu} \nabla_\sigma X^\rho \,, \tag{A.1a}$$

$$[\nabla_\mu, \nabla_\nu] \omega_\rho = R_{\mu\nu\rho}{}^\sigma \omega_\sigma - T^\sigma{}_{\mu\nu} \nabla_\sigma \omega_\sigma \,. \tag{A.1b}$$

---

[3]See [53, 68, 69] for recent work on Carroll symmetry in string theory.

This corresponds to the conventions

$$R_{\mu\nu\sigma}{}^{\rho} = -\partial_\mu\Gamma^\rho_{\nu\sigma} + \partial_\nu\Gamma^\rho_{\mu\sigma} - \Gamma^\rho_{\mu\lambda}\Gamma^\lambda_{\nu\sigma} + \Gamma^\rho_{\nu\lambda}\Gamma^\lambda_{\mu\sigma}\,, \tag{A.2}$$

$$T^\rho{}_{\mu\nu} = 2\Gamma^\rho_{[\mu\nu]}\,. \tag{A.3}$$

The Riemann tensor satisfies the Bianchi identity,

$$R_{[\mu\nu\sigma]}{}^{\rho} = T^\lambda{}_{[\mu\nu}T^\rho{}_{\sigma]\lambda} - \nabla_{[\mu}T^\rho{}_{\nu\sigma]}\,. \tag{A.4}$$

We define the Ricci tensor as the contraction $R_{\mu\nu} = R_{\mu\rho\nu}{}^{\rho}$. For connections where the trace of the connection coefficients is a total derivative $\Gamma^\sigma_{\mu\sigma} = \partial_\mu f$, one can show that the antisymmetric part of the Ricci tensor is given by

$$2R_{[\mu\nu]} = -2T^\lambda{}_{\sigma[\mu}T^\sigma{}_{\nu]\lambda} + T^\lambda{}_{\mu\nu}T^\sigma{}_{\lambda\sigma} + \nabla_\mu T^\sigma{}_{\nu\sigma} - \nabla_\nu T^\sigma{}_{\mu\sigma} + \nabla_\sigma T^\sigma{}_{\mu\nu}\,. \tag{A.5}$$

Lie derivatives can be written in terms of a general torsionful connections using

$$\begin{aligned}
\mathcal{L}_\xi X^{\mu_1\dots\mu_k}{}_{\nu_1\dots\nu_\ell} = {}&\xi^\lambda\nabla_\lambda X^{\mu_1\dots\mu_k}{}_{\nu_1\dots\nu_\ell} \\
&- \nabla_\lambda\xi^{\mu_1}X^{\lambda\mu_2\dots\mu_k}{}_{\nu_1\dots\nu_\ell} - \dots - \xi^\sigma T^{\mu_1}{}_{\sigma\lambda}X^{\lambda\mu_2\dots\mu_k}{}_{\nu_1\dots\nu_\ell} - \dots \\
&+ \nabla_{\nu_1}\xi^\lambda X^{\mu_1\dots\mu_k}{}_{\lambda\nu_2\dots\nu_\ell} + \dots + \xi^\sigma T^\lambda{}_{\sigma\nu_1}X^{\mu_1\dots\mu_k}{}_{\lambda\nu_2\dots\nu_\ell} + \dots\,,
\end{aligned} \tag{A.6}$$

which reduces to the familiar expression for vanishing torsion.

## A.2 Properties of the $\tilde\Gamma$ connection

Following the discussion in Appendix B, we consider the Carroll-compatible connection

$$\begin{aligned}
\tilde\Gamma^\rho_{\mu\nu} = {}&-v^\rho\partial_{(\mu}\tau_{\nu)} - v^\rho\tau_{(\mu}\mathcal{L}_v\tau_{\nu)} \\
&+ \frac{1}{2}h^{\rho\lambda}\left[\partial_\mu h_{\nu\lambda} + \partial_\nu h_{\lambda\mu} - \partial_\lambda h_{\mu\nu}\right] - h^{\rho\lambda}\tau_\nu K_{\mu\lambda}\,.
\end{aligned} \tag{A.7}$$

The corresponding covariant derivative of the Carroll metric variables gives

$$\tilde\nabla_\rho v^\mu = 0\,, \tag{A.8a}$$

$$\tilde\nabla_\mu\tau_\nu = \frac{1}{2}\tau_{\mu\nu} - v^\rho\tau_{\rho(\mu}\tau_{\nu)}\,, \tag{A.8b}$$

$$\tilde\nabla_\rho h_{\mu\nu} = 0\,, \tag{A.8c}$$

$$\tilde\nabla_\rho h^{\mu\nu} = v^{(\mu}h^{\nu)\sigma}(\delta^\gamma_\rho - \tau_\rho v^\gamma)\tau_{\gamma\sigma}\,, \tag{A.8d}$$

where $\tau_{\mu\nu} = 2\partial_{[\mu}\tau_{\nu]}$. This connection satisfies the following total derivative identities,

$$\partial_\mu(eX^\mu) = e(\tilde\nabla_\mu X^\mu + \tau_\mu X^\mu K)\,, \tag{A.9a}$$

$$\partial_\nu(eQ^{[\mu\nu]}) = e(\tilde\nabla_\nu Q^{[\mu\nu]} + \tau_\sigma K^\mu{}_\rho Q^{[\sigma\rho]} + K\tau_\nu Q^{[\mu\nu]})\,, \tag{A.9b}$$

where the measure $e$ is defined as

$$e = \sqrt{\det(\tau_\mu\tau_\nu + h_{\mu\nu})} = \det(\tau_\mu, e_\mu{}^a)\,. \tag{A.10}$$

The torsion of this connection is

$$\tilde T^\rho{}_{\mu\nu} = 2\tilde\Gamma^\rho_{[\mu\nu]} = 2h^{\rho\lambda}\tau_{[\mu}K_{\nu]\lambda}\,. \tag{A.11}$$

From the Carroll metric-compatibility (A.8a) and (A.8c), one can derive the following identities for the associated Riemann tensor,

$$\tilde{R}_{\mu\nu\sigma}{}^{\rho}v^{\sigma} = 0, \tag{A.12a}$$

$$\tilde{R}_{\mu\nu\sigma}{}^{\lambda}h_{\lambda\rho} + \tilde{R}_{\mu\nu\rho}{}^{\lambda}h_{\lambda\sigma} = 0. \tag{A.12b}$$

Finally, it can be shown that the trace of the connection is a total derivative

$$\tilde{\Gamma}^{\rho}_{\mu\rho} = \partial_{\mu}\log e, \tag{A.13}$$

which enables one to determine the antisymmetric part of the Ricci tensor,

$$\tilde{R}_{[\mu\nu]} = \tilde{\nabla}_{[\mu}(\tau_{\nu]}K) + \tilde{\nabla}_{\rho}(\tau_{[\mu}K^{\rho}{}_{\nu]}). \tag{A.14}$$

This implies in particular that the Ricci tensor is symmetric if the extrinsic curvature $K_{\mu\nu}$ vanishes, which is relevant in the 'magnetic' theory considered in the main text.

# B Frame derivation of Carroll-compatible connection

We now give a detailed derivation of the Carroll-compatible connection $\tilde{\Gamma}^{\rho}_{\mu\nu}$ that was introduced in Section 2.3. For this, we first translate the consequences of the 'pre-ultra-local' (PUL) parametrization of the metric to frames, following the analogous Galilean discussion in [77]. This frame perspective then gives us a concise approach to Carroll-compatibility, which can subsequently be translated back to the tangent bundle.

## B.1 Pre-ultra-local decomposition in frames

Recall that the Lorentzian metric $g_{\mu\nu}$ and its inverse can be described using a set of vielbeine and their inverses. These are one-forms and vector fields in the tangent bundle respectively, which can be written as

$$E^A = E_{\mu}{}^A dx^{\mu}, \qquad \Theta_A = \Theta^{\mu}{}_A \partial_{\mu}. \tag{B.1}$$

Here, the frame indices are $A = 0, 1, \cdots, d$, where $(d+1)$ is the spacetime dimension. The vielbeine can be chosen such that the metric and its inverse are given by

$$g_{\mu\nu} = \eta_{AB}E_{\mu}{}^A E_{\nu}{}^B, \qquad g^{\mu\nu} = \eta^{AB}\Theta^{\mu}{}_A \Theta^{\nu}{}_B. \tag{B.2}$$

In our conventions, the frame bundle metric $\eta_{AB} = \mathrm{diag}(-1, 1, \cdots, 1)$ and its inverse do not contain any explicit factors of $c$. The pre-ultra-local (PUL) decomposition (2.1) of the tangent bundle metric, which we repeat here for convenience,

$$g_{\mu\nu} = -c^2 T_{\mu}T_{\nu} + \Pi_{\mu\nu}, \qquad g^{\mu\nu} = -\frac{1}{c^2}V^{\mu}V^{\nu} + \Pi^{\mu\nu}, \tag{B.3}$$

then corresponds to the following decomposition of the frame metric,

$$\eta_{AB} = -t_A t_B + s_{AB}, \qquad \eta^{AB} = -t^A t^B + s^{AB}. \tag{B.4}$$

Here, $t_A t_B$ is a rank 1 matrix corresponding to the timelike direction, while the rank $d$ matrix $s_{AB}$ singles out the spacelike directions. Mirroring Equations (2.2) and (2.5), these tensors satisfy the following orthonormality and completeness relations,

$$t_A t^A = -1, \quad t_A s^{AB} = 0, \quad s_{AB}t^B = 0, \quad \delta^A_B = -t^A t_B + s^{AC}s_{CB}. \tag{B.5}$$

It is convenient to choose an orthonormal basis in the frame bundle such that

$$t_A = \delta_A^0, \quad t^A = -\delta_0^A, \quad s_{AB} = \delta_A^a \delta_B^b \delta_{ab}, \quad s^{AB} = \delta_a^A \delta_b^B \delta^{ab}, \tag{B.6}$$

which splits the indices $A = (0, a)$ in a time index 0 and $d$ space indices $a$. We can then use the Kronecker deltas $\delta_{ab}$ and $\delta^{ab}$ to raise and lower spacelike indices.

Next, we use these adapted coordinates to split the vielbeine (B.1) into spatial and temporal components, which allows us to define the PUL vielbeine as

$$t_A E_\mu{}^A = c\, T_\mu, \quad \delta_B^a E_\mu{}^B = \mathcal{E}_\mu{}^a, \quad t^A \Theta^\mu{}_A = -\frac{1}{c} V^\mu, \quad \delta_a^B \Theta^\mu{}_B = \Theta^\mu{}_a. \tag{B.7}$$

As a result, following the decomposition (B.4) of the frame bundle metric, the tangent bundle metric and its inverse can be written as

$$g_{\mu\nu} = (-t_A t_B + s_{AB}) E_\mu{}^A E_\nu{}^B = -c^2 T_\mu T_\nu + \delta_{ab} \mathcal{E}_\mu{}^a \mathcal{E}_\nu{}^b, \tag{B.8a}$$

$$g^{\mu\nu} = (-t^A t^B + s^{AB}) \Theta^\mu{}_A \Theta^\nu{}_B = -\frac{1}{c^2} V^\mu V^\nu + \delta^{ab} \Theta^\mu{}_a \Theta^\nu{}_b. \tag{B.8b}$$

With $\Pi_{\mu\nu} = \delta_{ab} \mathcal{E}_\mu{}^a \mathcal{E}_\nu{}^b$ and $\Pi^{\mu\nu} = \delta^{ab} \Theta^\mu{}_a \Theta^\nu{}_b$, this reproduces the decomposition (B.3).

On a Lorentzian manifold, the vielbeine transform under local Lorentz transformations, corresponding to the local Lorentz symmetry algebra of spacetime. Typically, such Lorentz transformations will not preserve the decompositions (B.3) or (B.4) of the tangent or frame bundle metric. Instead, these decompositions are adapted to the local Carroll transformations that appear at leading order in the small $c$ expansion, as we discussed in Section 2.2.

## B.2 Carroll-compatible connection

Recall that a tangent bundle connection $\Gamma^\rho_{\mu\nu}$ can be related to a frame bundle (or 'spin') connection $\Omega_\mu{}^A{}_B$ and vice versa using the relation

$$0 = \partial_\mu E_\nu{}^A - \Gamma^\rho_{\mu\nu} E_\rho{}^A + \Omega_\mu{}^A{}_B E_\nu{}^B, \tag{B.9}$$

which is also known as the vielbein postulate. We will now give a derivation of the Carroll connection introduced in Section 2 using frame language, which is more convenient for some of the manipulations.

Using the PUL decomposition of the frame bundle metric (B.4) and the vielbeine (B.7), we see that the Carroll metric compatibility from Equation (2.11) corresponds to

$$Dt^A = \Omega^A{}_B t^B = 0, \quad Ds_{AB} = -\Omega_A{}^C s_{CB} - \Omega_B{}^C s_{AC}, \tag{B.10}$$

which implies, using the decomposition $A = (0, a)$, that the following one-form connection components vanish,

$$\tilde{\Omega}^0{}_a = 0, \quad \tilde{\Omega}^0{}_0 = 0, \quad \tilde{\Omega}_{ab} + \tilde{\Omega}_{ba} = 0. \tag{B.11}$$

In the last equation, we have lowered the spatial index using the Kronecker delta corresponding to $s_{AB}$ in the orthonormal basis (B.6). As a result, we find that the components $\tilde{T}^A(\Theta_B, \Theta_C) = \tilde{T}_{\mu\nu}{}^A \Theta^\mu{}_B \Theta^\nu{}_C$ of the torsion two-form $\tilde{T}^A = dE^A + \tilde{\Omega}^A{}_B \wedge E^B$ are

$$\tilde{T}^0(\Theta_a, \Theta_b) = c\, d\tilde{T}(\Theta_a, \Theta_b) + \tilde{\Omega}^0{}_b(\Theta_a) - \tilde{\Omega}^0{}_a(\Theta_b), \tag{B.12a}$$

$$\tilde{T}^0(\Theta_a, V) = c\, d\tilde{T}(\Theta_a, V) - \tilde{\Omega}^0{}_a(V), \tag{B.12b}$$

$$\tilde{T}_a(\Theta_b, \Theta_c) = d\mathcal{E}_a(\Theta_b, \Theta_c) + \tilde{\Omega}_{ac}(\Theta_b) - \tilde{\Omega}_{ab}(\Theta_c), \tag{B.12c}$$

$$\tilde{T}_a(\Theta_b, V) = d\mathcal{E}_a(\Theta_b, V) - \tilde{\Omega}_{ab}(V). \tag{B.12d}$$

By symmetrizing Equation (B.12d), we see that part of the torsion is independent of our choice of connection,

$$\tilde{T}_{(a}(\Theta_{b)}, V) = d\mathcal{E}_{(a}(\Theta_{b)}, V). \tag{B.13}$$

In other words, these components of the torsion are *intrinsic* to any Carroll-compatible connection satisfying the requirement (B.11), in agreement with the recent classification in [86]. No choice of Carroll-compatible connection can set them to zero, and manually setting them to zero would impose a constraint on the spacetime geometry. Therefore, for an arbitrary Carroll manifold, the constraints

$$T^0(\Theta_a, \Theta_b) = 0, \quad T^0(\Theta_a, V) = 0, \quad T_a(\Theta_b, \Theta_c) = 0, \quad T_{[a}(\Theta_{b]}, V) = 0, \tag{B.14}$$

are the strongest torsion constraints we can impose on our Carroll-compatible connection.

Note that the constraints (B.14) and the compatibility conditions (B.11) do not determine the connection uniquely, since they do not allow us to solve for the symmetrized components $\tilde{\Omega}^0{}_{(a}(\Theta_{b)})$. These components of the connection would enter in the covariant derivative of the tensor $t_A$, which is also not invariant under Carroll boosts. To simplify this covariant derivative as much as possible in a given boost frame, we set these symmetrized components to zero. Solving Equation (B.12) then gives us

$$\tilde{\Omega}^0{}_a(\Theta_b) = \frac{c}{2} dT(\Theta_a, \Theta_b), \tag{B.15a}$$

$$\tilde{\Omega}^0{}_a(V) = c\, dT(\Theta_a, V), \tag{B.15b}$$

$$\tilde{\Omega}_{ab}(\Theta_c) = \frac{1}{2}\left[d\mathcal{E}_a(\Theta_b, \Theta_c) + d\mathcal{E}_b(\Theta_c, \Theta_a) - d\mathcal{E}_c(\Theta_a, \Theta_b)\right], \tag{B.15c}$$

$$\tilde{\Omega}_{ab}(V) = d\mathcal{E}_a(\Theta_b, V). \tag{B.15d}$$

This fixes our preferred connection in the frame bundle. We can then use the vielbein postulate (B.9) to transform this connection to the tangent bundle,

$$\tilde{C}^\rho_{\mu\nu} = \Theta^\rho{}_A\left(\partial_\mu E_\nu{}^A + \tilde{\Omega}_\mu{}^A{}_B E_\nu{}^B\right) \tag{B.16}$$

$$= -V^\rho\left(\partial_\mu T_\nu + \frac{1}{c}\tilde{\Omega}_\mu{}^0{}_a \mathcal{E}_\nu{}^a\right) + \Theta^\rho{}_a\left(\partial_\mu \mathcal{E}_\nu{}^a + \tilde{\Omega}_\mu{}^a{}_b \mathcal{E}_\nu{}^b\right) \tag{B.17}$$

$$= -V^\rho \partial_{(\mu} T_{\nu)} - V^\rho T_{(\mu}\mathcal{L}_V T_{\nu)} \tag{B.18}$$

$$+ \frac{1}{2}\Pi^{\rho\lambda}\left[\partial_\mu \Pi_{\nu\lambda} + \partial_\nu \Pi_{\lambda\mu} - \partial_\lambda \Pi_{\mu\nu}\right] - \Pi^{\rho\lambda} T_\nu \mathcal{K}_{\mu\lambda},$$

which corresponds to the Carroll-compatible PUL connection in Equation (2.13). The torsion of this connection, which is constructed to be have only intrinsic torsion corresponding to (B.13), is given by

$$2\tilde{C}^\rho_{[\mu\nu]} = 2\Pi^{\rho\lambda} T_{[\mu}\mathcal{K}_{\nu]\lambda}. \tag{B.19}$$

This reflects the result that the intrinsic torsion of a Carroll-compatible connection is determined by the extrinsic curvature $K_{\mu\nu}$ [86]. Following Theorem 10 in [86], such connections can be classified in i) vanishing $K_{\mu\nu}$, ii) traceless $K_{\mu\nu}$, iii) $K_{\mu\nu} = f h_{\mu\nu}$, or iv) none of the above. This mirrors the similar classification of Newton–Cartan connections in torsionless ($d\tau = 0$), twistless-torsional ($\tau \wedge d\tau = 0$) or general torsion.

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
