# Peer review of "Carroll Expansion of General Relativity"

_SciPost Physics, doi:SciPost Phys. 13, 055 (2022)_

## Round 2 · Referee Report · Luca Ciambelli (Referee 1) · 2022-3-8

Strengths

1- Clear and detailed, easy to follow 2- Various appendixes with further details and explanations

Weaknesses

1- Difficult to disentangled new original results from already established results 2- Some more comments could have been made about the general interest of the topic

Report

In this paper, the authors perform the $c \rightarrow 0$ expansion of the Lagrangian of general relativity. They show how the leading and next-to leading order are related to the electric and magnetic Carroll theories that already appeared in the literature. They then, exporting tools of the 3+1 decomposition of general relativity, find solutions to the evolution equations of the leading order Carroll theory. They conclude discussing the Carroll limit of the Schwarzschild solution, and how to include the cosmological constant in the derivation.

In my view, the content of the paper is very interesting, and the ideas presented worth pursuing. It sits well in an important list of manuscripts devoted to new fascinating ideas in this discipline. Nonetheless, this paper has some minor flaws, and the presentation of the results could be improved, by clearly stating what is original and what is an extension or (better) rewriting of already established results. I collect these comments in the ''requested changes'' section below.

In conclusion, I believe that the ideas in the paper are sufficiently interesting to warrant publication in SciPost. Therefore, I will be glad to reconsider the paper once these minor flaws are addressed, and the clarity in discerning new results improved.

Requested changes

I start with general comments:

1- While I am personally convinced that this field is interesting and worth pursuing, I believe the authors could provide some more general thoughts as why we should do so. What is the interest of performing such limit? Are the authors expecting to be able to say more about general relativity and quantum gravity, or do they rather expect to find Carrollian realization of gravity in some physical systems? I believe a discussion about long-term objectives of these queries could improve the manuscript.

2- I believe the distinction between known results and new, original, results could be sharpen. Is the general goal to rewrite the Hamiltonian analysis of Henneaux and Salgado-Rebolledo in the Lagrangian formalism? Or is it rather to find explicit solutions to the leading order limit, and are these new original solutions? A clearer distinction could highlight better the authors' new results.

3- Equations (2.4a)-(2.4c) are crucial in the following. There is a (natural) choice implicitly made there, which is to start the expansion at order $c^0$. I suggest to spend considerably more time commenting on this choice. Although natural, is it unique? What would change by making a different choice? Are the scaling in $c$ dictated by some physical assumptions?

On top of these two broader comments, there are a number of more aestethic-related issues and typos I would like to point out:

1- In the first paragraph of the introduction, after "applications in astrophysics and cosmology", the authors should point out some references. 2- In the third-last paragraph of the introduction, the authors should amend the sentence "This Carroll geometry corresponds was also considered in" 3- At the end of the introduction, a precise road map of the paper could be added 4- In (2.7b), the first equation has a wrong index structure (I believe it should should be $V^\mu$). 5- In (2.9), the third equation has a wrong index structure (I believe it should be $v^\mu$). 6- Is $\delta_\lambda v^\mu=0$ compatible with $\delta_\lambda (v^\mu\tau_\mu)=\delta_\lambda (-1)=0$? Or is the latter property untrue? 7- In (2.10), the second equation has a wrong index structure (I believe it should be $h^{\mu\nu}\lambda_\nu$). 8- How the discussion of section 2.5 is related to the fibre bundle interpretation proposed in Phys.Rev.D 100 (2019) 4, 046010 ? 9- In (2.30), it is true that one can use Carroll boosts to set $b_i$ to zero, but it seems that the authors want to keep this boost symmetry available (residual), so why freezing it? 10- Is the Carroll compatible connection introduced here the same as the one appearing in Arxiv 1802.06809 (JHEP 07 (2018) 165)? 11- In (3.5), the authors use as variables $v^\mu$ and $h^{\mu\nu}$. However, as they show previously, the contravariant $h^{\mu\nu}$ is not invariant under Carroll boosts. It seems therefore more reasonable to use $h_{\mu\nu}$ instead. Could they comment on this point? 12- This last comment could have important consequences for the rest of their analysis: are the resulting Lagrangians and equations of motion invariant under Carroll boosts and, if not, is there an issue? 13- Similarly, in (3.8), is it not better for the next-to leading order to use the variable $\Phi_{\mu\nu}$, instead of its contravariant dual? 14- Is (4.3) a new original result, presented here for the first time? It seems the authors claim so, although I find it surprising that such a solution has not appeared before in the context of 3+1 split.

  • validity: -
  • significance: -
  • originality: -
  • clarity: -
  • formatting: -
  • grammar: -

Author:  Gerben Oling  on 2022-03-30  [id 2338]

(in reply to Report 1 by Luca Ciambelli on 2022-03-08)

We thank the referee for their detailed report and their constructive comments.

The three general comments will be addressed in an upcoming revision by clarifying and adding additional emphasis to our Introduction and Discussion sections as well as the introductory texts at the start of each section. Briefly, our response is as follows:

  1. Our motivations for studying the Carroll expansion of general relativity lie in the covariant description of ultra-local gravity effects that it provides, zooming in on a particular tractable yet nontrivial subsector of the dynamics of the full relativistic theory, as exemplified by the solutions we find in this paper. Additionally, the tools we develop here (in particular the relation to the 3+1 decomposition) are also expected to be useful for the non-relativistic Galilean expansion.
  2. Our main result is the development of a systematic Carroll expansion of general relativity. As an example of this general formalism, we show that it can easily provide a Lagrangian description for the 'electric' and 'magnetic' Hamiltonian limits described by Henneaux and Salgado. These theories are special subcases arising from our general Carroll expansion, corresponding to the leading-order and a truncation of the next-to-leading-order theories, respectively. Our identifications of these theories as a subset of our general framework serves both to make contact with the existing literature and to identify particular tractable subsectors where first examples of explicit solutions can be obtained.
  3. It is true that the specification of our expansion in Equations (2.4a)-(2.4c) contains important physical information. However, this is not so much the fact that this expansion starts at order $c^0$ (since we have already split off the factors of $c$ dictated by dimensional analysis in (2.1) or (2.3)), but rather the assumption that these vielbeine admit an analytic expansion in powers of $c^2$. As we already explained in Footnote 2, this restricts the class of metrics and in particular the coordinates in which our expansion can be performed. In particular, $c$-dependent symmetry transformations of the relativistic theory such as local boosts will naturally lead to different starting points for our covariant expansions. Similar considerations enter in the Galilean non-relativistic expansion of general relativity.

On the more specific comments:

  1. Additional references will be added in a new version.
  2. Additional references will be added in a new version.
  3. We will add a 'road map' of the paper at the end of the introduction in a new version.
  4. Correct, this will be fixed in a new version.
  5. Correct, this will be fixed in a new version.
  6. The boost-invariance $\delta_\lambda v^\mu = 0$ is indeed compatible with the relation $v^\mu \tau_\mu = 0$, since $\tau_\mu$ transforms under boosts with a spatial vector which vanishes upon contraction with $v^\mu$. We will add a comment emphasizing the latter in a new version.
  7. Correct, this will be fixed in a new version.
  8. The fiber bundle description of Carroll manifolds in the paper quoted by the referee can be obtained from our description of Carroll geometry by identifying the fibers with the integral curves of our $v^\mu$ vector field. The base space of this fiber bundle then corresponds to our notion of the 'spatial submanifold'. However, as remarked in our Section 2.5, one can typically not define spatial hypersurfaces in a Carroll boost-invariant way since $\tau_\mu$ transforms. This ambiguity is parametrized by the spatial components $b_i$ of $\tau_\mu$ in our Equation (2.30). On the fiber bundle side, it is reflected by the choice in Ehresmann connection $b_i$. (Likewise, the quantity $\omega$ on the fiber bundle side is related to our 'lapse' parameter $\alpha$.) We will add a footnote discussing this relation in a new version.
  9. Following up on this point, in a boost-fixed frame, one can use the projector operators (2.32) to separate spatial and timelike components of tensors. In the fiber bundle language, this would correspond to fixing a particular Ehresmann connection, which likewise splits its tangent spaces in a sum of vertical (timelike) and horizontal (spacelike) subspaces. In particular one can use Carroll boosts to set $b_i=0$, so that $\tau_\mu$ is purely temporal. As a result, it satisfies the Frobenius condition $\tau\wedge d\tau=0$, and the spatial subspaces designated by the spatial projector can be integrated to a spatial hypersurface in the full spacetime manifold. (In the fiber bundle language, this means that the Ehresmann connection vanishes and hence the fiber structure is trivial. Our possibly non-constant lapse is mapped to the separate variable $\omega$. We will also add a footnote on this in a new version.) The main point of our Section 2.5 is that working in such a boost-fixed frame with a well-defined spatial foliation allows us to describe the pullback of various tensors to these spatial surfaces. In particular in the context of the subsector of the NLO theory corresponding to the 'magnetic' limit, it is useful to have access to the spatial Riemann tensor and its contractions. However, at the end of the day, while they may be obtained in a particular convenient boost frame, our boost-covariant results (and in particular the solutions discussed in Section 4) can be considered in any boost frame.
  10. The Carroll-compatible connection in our (2.14) appears to be closely related to the connections introduced for the fiber bundle perspective in (3.8) and (3.12) of 1802.06809. However, the language is sufficiently different that a comparison is not straightforward. We hope to clarify this relation in future work.
  11. Even though $h^{\mu\nu}$ transforms non-trivially under boost transformations and $h_{\mu\nu}$ does not, we consider variations with respect to $v^\mu$ and $h^{\mu\nu}$ instead of deriving equations because we want to keep the Carroll boost gauge symmetry explicit. The resulting equations of motion then transform covariantly under boosts. Relatedly, note that one cannot solve $(\tau_\mu, h^{\mu\nu})$ from $(v^\mu, h_{\mu\nu})$ precisely because of the ambiguities due to boosts that are present in the former pair but absent from the latter. In contrast, the pairs $(v^\mu, h^{\mu\nu})$ and $(\tau_\mu, h_{\mu\nu})$ can be expressed in terms of each other, and either choice therefore fully specifies the Carroll metric data including its boost frame.
  12. Following up on the previous points, the actions that we obtain at each order are invariant under boosts, whereas the equations of motions are covariant, in analogy to the diffeomorphism gauge symmetry. Furthermore, boost transformations lead to Noether/Ward identities satisfied by the equations of motion. For example, the nontrivial (timelike) transformation of $\delta h^{\mu\nu}$ under boosts implies that the corresponding equation of motion is purely spatial.
  13. Our reason for varying with respect to $\Phi^{\mu\nu}$ instead of $\Phi_{\mu\nu}$ at NLO is the same as the difference between varying with respect to $h^{\mu\nu}$ and $h_{\mu\nu}$ discussed in point 11. In most of our paper, we consider only a truncated sector of the NLO dynamics where these fields are turned off, but if they would be present, their transformations would imply similar Noether/Ward identities on the corresponding equations of motion.
  14. As mentioned in the text the solution (4.3) hinges on the 'ultra-local' nature of the leading-order equations which decouples spatially separated points. Such a simplification of the evolution equation is therefore typically not present in the full relativistic evolution equation. However, the LO/electric Carroll theory does appear to be closely related to the Belinsky-Khalatnikov-Lifshitz limit of general relativity, which we will point out in a new version.

---

## Round 2 · Referee Report · Anonymous (Referee 2) · 2022-3-30

Strengths

  1. Potentially important development in the field.
  2. Generally well written.
  3. Contains relevant details in appendices.

Weaknesses

  1. Could do with some more engagement with the uninitiated reader as to why this is an important limit to consider.
  2. Since the post-Newtonian expansion is mentioned and compared with at various points, an appendix on some of the important features is something the authors should look to put in to make the paper more self-contained.

Report

In the paper under review, the authors consider the Carrollian expansion of gravity, where the speed of light is taken to zero. This is the diametrically opposite limit to the Galilean expansion where $c \to \infty$. This limit has become very fashionable of late due to potential connections to holography in asymptotically flat spacetimes, links to high energy strings, and other intriguing avenues like cosmology and dark energy, and the theory of fractons.

The authors study General Relativity around the Carroll point in a spirit similar to the Post Newtonian (PN) approximation of GR, in what they call pre-ultra local variables. The leading order and next-to-leading order (NLO) expansion of the action yields the so called Electric and Magnetic versions of the Carroll limit of GR. Interestingly, the leading order equations from the Electric theory can be solved analytically. The notion of mass enters the NLO term in the action and the authors study solutions which are the Carroll equivalent of the Schwarzschild black hole and also discuss how to incorporate non-zero cosmological constants.

The article presents a potentially important direction in this emerging field and I would recommend publication once some of my (mostly minor) questions and comments are addressed.

Requested changes

  1. From the introduction, it is not clear why the $c\to0$ of GR is of interest. It is an analogue of the PN approximation and possibly a systematic expansion, but what is the physics that this expansion captures? The authors say that this provides novel insights into geometry and gravitational dynamics. It would be good if the authors expanded on this substantially.

  2. Since there are constant references to the non-relativistic analogue throughout the paper, I would suggest that the authors put together an appendix which contains some details of the PN approximation that is of direct interest for this paper. This would help in the reading of the paper and make it more self contained.

  3. The electric and magnetic theories differ quite substantially, which is of course evident from the actions that the authors write down. But a priori is there any reason to expect such departures? e.g. is it a priori obvious, or clear in hindsight, that the leading term in the action would not contain a notion of mass or that a cosmological constant would only be admissible in the NLO term?

  4. There is a typo in the expression of $\Phi^{\mu\nu}$ below Eq (2.4).

  5. Another typo: in Eq (2.9) the second equation should be $v^\mu$ instead of $v_\mu$

  6. The expression for $R^{(0)}_{\mu\nu}$ does not contain any $S^2$ terms, which naively have to be there. Is it easy to see why this is happening?

  7. In and around Eq (3.5): the fields in the action seem to be $v^\mu$ and $h^{\mu\nu}$. I would have expected the authors to use the Carroll boost invariant $v^\mu$ and $h_{\mu\nu}$. Why is the above choice consistent? Why does working with the second one not give the correct formulation? I guess the questions are even more pertinent given the explicit discussion the authors have in Sec 2.5.

  8. The authors describe a "Schwarzschild" solution. What does this solution mean for black holes? The $c\to0$ limit usually means localisation on some null hypersurface. Does this mean that the solution is the leading term in an expansion around the horizon of the original Schwarzschild? Do the magnetic theory somehow capture the blackhole horizon dynamics?

  • validity: high
  • significance: high
  • originality: good
  • clarity: good
  • formatting: excellent
  • grammar: excellent

Author:  Gerben Oling  on 2022-07-04  [id 2628]

(in reply to Report 2 on 2022-03-30)
Category:
remark
answer to question

We thank the referee for their useful remarks and apologize for our late reply.

First, as a minor technical point, we should emphasize that the magnetic Carroll limit of GR is only a subset of the full NLO theory we construct in our expansion. Following the comments of Report 1, we have clarified this further in our Introduction. Next, on the requested changes in Report 2:

  1. While it indeed serves as a major motivation for our work, it is important to note that the precise identification between the Galilean Newton--Cartan expansion of GR and the post-Newtonian expansion is still a subject of ongoing work. To clarify this, we have added a comment to this extent to our Discussion.
  2. It is true that there are technical similarities between the Carroll and Galilei expansion. However, we feel that any helpful summary of the Galilean expansion would be too long to be appropriate to include here. Instead, to aid the reader we have included more detailed references to specific sections and equations of 2001.10277.
  3. It is an interesting question whether there are any underlying reasons for the physical differences between the magnetic and electric limits. Apart from their different technical origins, the fact that the electric limit only depends on the intrinsic Carroll data $v^\mu$ naturally makes it more tuned for the intuitive Carroll behavior described by independent worldlines. In the magnetic case, since the time evolution $L_v h_{\mu\nu}$ of the spatial metric vanishes due to the constraint $K_{\mu\nu}=0$, it deals with the Carroll symmetry in a different way. Physically, perhaps the most straightforward interpretation of this dichotomy is as the two distinct ways to solve the Ward idenitity for Carroll boosts, which constrains scalar two-point functions $\langle \phi(t,x) \phi(0,0) \rangle$ to be $f(t)\delta(x)$ or $g(x)$, where the former case is 'electric' and the latter case is 'magnetic'. However, since the generalization of this statement to spinning propagators has not yet been developed, we have refrained from commenting on this in the present draft. Furthermore, we do not know of an argument that would make it clear why mass arises in the 'magnetic' and not in the 'electric' sector of the Carroll expansion of gravity.
  4. This typo has been fixed in v3, in fact the previous expression $\Phi^{\mu\nu} = \delta^{ab} \pi^\mu{}_a \pi^\nu{}_b$ was wrong, it should instead be $\Phi^{\mu\nu} = \delta^{ab} \left(\theta^\mu{}_a \pi^\nu{}_b + \pi^\mu{}_a \theta^\nu{}_b\right)$.
  5. The typo (in the second equality of Equation (2.9)) has been fixed in v3.
  6. As far as we know, there is deep reason for why the $S^2$ terms in the PUL expansion of the Levi-Civita Ricci tensor at $c^0$ vanish, it essentially arises due to a contraction $T_\nu\Pi^{\nu\sigma}=0$ contraction. We have pointed out this fact in a small comment below Equation (2.25).
  7. This comment corresponds to the specific comment 11 of Report 1, which we have now addressed with an extra paragraph below Equation (3.6).
  8. This is an interesting question. Based on its relation to the BKL limit, the electric limit appears to focus on the near-singularity limit of the Schwarzschild metric, rather than the near-horizon limit. On the other hand, it appears that the magnetic limit describes the asymptotic region of a black hole in Schwarzschild-type coordinates. (For this, we should also credit conversations with Stefan Vandoren at the 2022 Carroll Workshop in Vienna.) We would like to explore this connection in the future.

---

## Round 3 · Referee Report · Anonymous (Referee 2) · 2022-7-17

Report

The authors have addressed my questions and comments. I now recommend the article for publication.

---

## Round 3 · Referee Report · Luca Ciambelli (Referee 1) · 2022-7-18

Report

The authors carefully addressed all my points in a systematic way.

There is still a minor detail (that I will not need to review) that I would like to point at, in the amended version. The discussion below eq. (2.30) is unclear. First, I do not understand the sentence "The local Carroll boost transformations (2.9) act by shifting bi → bi + λi, corresponding to the choice of Ehresmann connection." Perhaps the authors meant "corresponding to a different the choice of Ehresmann connection." ? In the sentence after, there is a closing round parenthesis ")" that I believe is a typo. In general, this paragraph could be amended and explained better.

I am glad to recommend this paper for publication in SciPost.

---

## Round 3 · Author Response

We have posted our response to each report in an individual reply. In the provided form we have listed some final changes that were not (fully) detailed in these individual replies.

---

## Round 3 · List of Changes

First, we have amended our Introduction with a paragraph detailing our precise physical and technical motivations for studying the Carroll expansion of general relativity. These points, which were previously mentioned only later on in the introduction, are now collected in the fourth and fifth paragraph of our Introduction. Additionally, in the ninth paragraph of the Introduction, we have clarified the relation of our expansion to the electric and magnetic limits, emphasizing that the latter are only special cases of our more general result. Furthermore, we have added a roadmap of our paper at the end of the Introduction, as requested in Report 1.

To address the third general comment in Report 1, we have expanded a previous footnote into a paragraph below Equation (2.4), explaining the assumptions underlying that expansion. We have expanded the first, second, third and last paragraph of Section 2.5 to address the specific comments 8 and 9 of Report 1. Additionally, we have added a paragraph below (3.6), addressing both the specific comments 11 and 12 from Report 1 and comment 7 from Report 2. Finally, we have fixed several other typos (including the one pointed out in the specific comment 2 of Report 1).

---

## Round 4 · Author Response

We thank both referees for their second report. We have clarified the point raised in Report 2 and we have rewritten the relevant paragraph.

---

## Round 4 · List of Changes

The paragraph directly below Equation (2.30) has been rewritten.

---

## Editorial Decision

published